# Interspecific differences in the effects of masking and distraction on anti-predator behavior in suburban anthropogenic noise

Trina L. Chou[1]☯, Anjali Krishna[2]☯, Mark Fossesca[1]‡, Avani Desai[2]‡, Julia Goldberg[2], Sophie Jones[2], Morgan Stephens[2], Benjamin M. Basile[3], Megan D. Gall[1,2]*

1 Neuroscience and Behavior Program, Vassar College, Poughkeepsie, NY, United States of America, 2 Biology Department, Vassar College, Poughkeepsie, NY, United States of America, 3 Department of Psychology, Dickinson College, Carlisle, PA, United States of America

☯ These authors contributed equally to this work.
‡ MF and AD also contributed equally to this work.
* megall@vassar.edu

**Data Availability Statement:** All data are available as supporting information files.

**Funding:** Funding for this project was provide by the Vassar Undergraduate Research Summer

## Abstract

Predation is a common threat to animal survival. The detection of predators or anti-predator communication signals can be disrupted by anthropogenic noise; however, the mechanism by which responses are affected is unclear. Masking and distraction are the two hypotheses that have emerged as likely explanations for changes in behavior in noise. Masking occurs when the signal and noise fall within the same sensory domain; noise overlapping the energy in the signal reduces signal detection. Distraction can occur when noise in any sensory domain contributes to a greater cognitive load, thereby reducing signal detection. Here, we used a repeated measures field experiment to determine the relative contributions of masking and distraction in mediating reduced anti-predator responses in noise. We recorded the approaches and vocalizations of black-capped chickadees (*Poecile atricapillus*), tufted titmice (*Baeolophus bicolor*), and white-breasted nuthatches (*Sitta carolinensis*) to both visual and acoustic cues of predator presence, either with or without simultaneous exposure to anthropogenic noise. Titmice increased their calling to both visual and acoustic cues of predator presence. However, there was no significant effect of noise on the calling responses of titmice regardless of stimulus modality. Noise appeared to produce a distraction effect in chickadees; however, this effect was small, suggesting that chickadees may be relatively unaffected by low levels of anthropogenic noise in suburban environments. White-breasted nuthatch calling behavior was affected by the interaction of the modality of the predator stimulus and the noise condition. Nuthatches had a delayed response to the predator presentations, with a greater calling rate following the presentation of the acoustic stimulus in quiet compared to the presentation of the acoustic stimulus in noise. However, there was no difference in calling rate between the quiet and noise conditions for the visual stimulus. Together this suggests that even moderate levels of noise have some masking effect for white-breasted nuthatches. We suggest that the mechanisms through which noise influences anti-predator behavior may depend on the social roles, foraging ecology and auditory capabilities of each species.

Institute, the Vassar College Biology Department, and the Asprey Center for Collaborative Approaches to Science. The funders had no role in study design, data collection and analysis, decision to publish, or preparation of the manuscript.

**Competing interests:** The authors have declared that no competing interests exist.

## Introduction

Avoiding predation is critical to the survival of many animals [1, 2]. Detecting a predator while it is not actively hunting can prevent an animal from becoming the subject of an attack [1, 2]. Such detection can occur in many sensory domains [2], and may come from direct observation of a predator (i.e. personal information [3–6]), or indirectly from observation of conspecific or heterospecific behavior (i.e. public information; [6–9]). In some cases, animals have evolved anti-predator communication signals such as mobbing calls, alarm calls and alarm pheromones to alert conspecifics to predation threats [1, 2].

Regardless of whether information about predators comes from direct cues, indirect cues, or communication signals, successful evasion requires animals to distinguish information from a complex sensory background [10–12]. Information about predators may not be successfully detected if there is sufficient noise in the system to either mask relevant signals [11] or increase the cognitive load of the animals to the point where they can no longer effectively process those signals, which may make detecting predators more difficult even if the sensory information is not directly masked [5, 13, 14]. This may be particularly true when background noise is fluctuating and unpredictable [15].

Anthropogenic noise is a common source of noise in urban and suburban environments. Anthropogenic noise has been linked to decreased predator detection in a variety of species [4, 11, 13, 16, 17]. Additionally, anthropogenic noise can reduce mobbing responses to anti-predator communication signals [18, 19]. Most of this work has used temporally-stable (i.e., no amplitude fluctuations) spectrally-filtered white noise [4, 13, 20]. White noise mimics some, but not all types of anthropogenic noise. In some cases, such as busy highways, noise can be both very loud (upwards of 70 dBA at 200 meters) and relatively temporally-stable [21]. However, in suburban areas, where the volume and speed of traffic is lower, noise levels are typically lower in amplitude with greater temporal variability [22]. Temporally-stable noise produces consistent levels of stimulus masking, but can also be easier to habituate to, causing relatively low levels of distraction [23]. On the other hand, temporally-fluctuating noise may allow sporadic relief from masking (e.g., via dip-listening), but is much more difficult to habituate to, and therefore may cause relatively high levels of cognitive distraction [24, 25], but see [26].

Although there has been significant work on the effects of anthropogenic noise on reproductive communication signals, and more recently on anti-predator communication, the mechanism by which anthropogenic noise decreases behavioral responses to stimuli is not yet clear. Reduced behavioral responses are often assumed to be the result of masking, in which noise overlaps the stimulus of interest [27–31]. Masking may be the predominant mechanism in studies that use spectrally-filtered white noise as a stand-in for anthropogenic noise. However, noise may also reduce behavioral responses via distraction, in which noise competes with ecologically relevant stimuli for attentional, rather than sensory space, resulting in a cognitive deficit [13, 32, 33]. Distraction may be particularly important in temporally-fluctuating noise, even when the noise would be expected to cause relatively little masking. While masking is confined to one sensory modality, distraction can occur both within and across sensory modalities. For example, acoustic noise cannot mask a visual signal, but it could distract from it [15]. The masking and distraction hypotheses are not mutually exclusive and may both play a role to varying degrees depending on the stimuli, the structure of the noise, and the experience of the animals.

Here, we investigated (1) whether temporally-fluctuating suburban noise is sufficient to alter the behavioral responses of black-capped chickadees (*Poecile atricapillus*, hereafter

chickadees), tufted titmice (*Baeolophus bicolor*, hereafter titmice), and white-breasted nuthatches (*Sitta carolinensis*, hereafter nuthatches) to predator cues and (2) the relative contributions of the two proposed mechanisms, masking and distraction, to the effect of anthropogenic noise on anti-predator behavior. These three species form mixed species flocks during the non-breeding season, engage in predator mobbing together, and are known to respond to the mobbing calls of chickadees [34, 35]. Chickadees and titmice both produce *chick-a-dee* calls and nuthatches produce *quank* calls in anti-predator contexts [35–37]. The *chick-a-dee* calls of chickadees and titmice contain a number of different note types that allow for flexibility in the use of the call. The number of D notes in a call is thought to encode predator attributes, such as size [37, 38]. *Quank* calls and the number of quank notes are also thought to be associated with increased excitement or potentially predator risk [36], but see [39].

We presented mixed-species flocks with either an acoustic cue of predator presence (playback of chickadee *chick-a-dee* mobbing calls) or a visual cue of predator presence (taxidermy mount of a Northern saw-whet owl, *Aegolius acadicus*). Predator-related stimuli were presented alone or with playback of traffic noise. If masking is primarily responsible for changes in anti-predator behavior, we would predict a decrease in the response to stimuli presented in the acoustic modality (mobbing calls), but not visual modality (the taxidermy mount). If distraction is primarily responsible for changes in anti-predator behavior, we would predict a decrease in the response to predator-related stimuli presented in both sensory modalities. We used a two-by-two counterbalanced repeated measures design to test these hypotheses. Each trial consisted of a pre-stimulus period to establish a behavioral baseline, a during-stimulus period to measure the effect of the independent variables, and a post stimulus period to capture extended effects of the presented cue (hereafter: pre, during, and post). We used the number of approaches to the stimulus, as well as the anti-predator vocal responses (number of calls and number of D or quank notes, [35–37]) of the three species as response variables.

## Methods

### Experimental design

We measured the anti-predator responses of chickadees, tufted titmice, and white-breasted nuthatches at 14 locations on the Vassar Farm and Ecological Preserve in Poughkeepsie, NY (41.66753425 N, 73.89584266 W) that were all exposed to sporadic, low levels of anthropogenic noise from nearby suburban streets. Each location had an elevated feeding platform which was baited with black oil sunflower seeds twice per week starting approximately two weeks before the beginning of data collection and continuing throughout the data collection period to encourage consistent bird activity (as in [18]). Platforms were at least 0.4 km apart and during our ten years of banding and subsequent observations we have never encountered individuals at more than one platform location in a given year. A total of 56 trials were conducted between 0800 and 1200 hours from November 13, 2021 to April 10, 2022. We counterbalanced noise type and predator stimulus modality in the trials for each platform across the experimental period. Each trial type (one of four combinations of the predator cue and noise stimuli) was performed once at each location, resulting in a total of 4 trials per platform with at least 7 days and a mean of 24 days between trials at a given platform. Experiments were never conducted at geographically adjacent platforms on the same day. All methods were approved under Vassar IACUC #20-06B and birds were banded under NY state (#260) and federal (23873) banding permits. No other permits were required for this work. Access to the field site was authorized by Vassar College.

## Stimuli, calibration and playback equipment

The Northern saw-whet owl stimulus was created from a skin obtained from the local population of saw-whet owls and commercially available saw-whet owl taxidermy forms and eyes. Northern saw-whet owls are natural predators of small birds and are viewed as high-threat predators by black-capped chickadees [37]. Chickadees produce vocalizations that encode a high degree of threat to saw-whet owls [37] and show higher neural activation to saw-whet directed vocalizations than to vocalizations directed towards lower threat predators [40]. We had only one predator mount to present and therefore have some level of pseudoreplication. However, given the robust responses to saw-whet owls in previous work and our robust responses here (in preliminary investigations, we found birds would respond even to a wooden saw-whet owl with inappropriate dimensions), we expect that the data gathered from this one model would generalize to other saw-whet owl models. Supporting this, recent publications on parids that have included multiple mounts (2–5 exemplars) have either not evaluated the effects of mount identity on behavioral responses [6, 41], found inconsistent effects [42], or found no effects [17, 43].

The acoustic cue of predator presence consisted of chickadee *chick-a-dee* calls with 3–4 D notes that were arranged in 22 or 23 bouts. Each bout contained four *chick-a-dee* calls, with a mean inter-call interval of 3.5 s and a mean interbout interval of 10 s [44]. The five stereotypical *chick-a-dee* calls used for the stimulus files were selected from recordings with minimal background noise (16-bit wave, sampling rate = 44.1 kHz) made at the Vassar Farm and Ecological Preserve in 2017 using a Marantz PMD661 field recorder, a Sennheiser ME66 microphone capsule equipped with a Rycote softie windscreen, and a K6 power unit. Calls of this type are in line with those produced to Northern saw-whet owls in previous work [37, 44] and all three target species respond to these calls [34].

The anthropogenic noise we used in our trials was recorded from five locations on the main road that abuts the Vassar College campus (Raymond Avenue; speed limit 30 mph) using a Marantz field recorder (PMD661 MKII) and an omnidirectional microphone (Sennheiser ME62 capsule, K6 power unit) with a Rycote softie windscreen. Each of the five noise files was trimmed to 10 minutes in Audacity 3.0 and scaled to the same RMS amplitude in PRAAT. Noise had peak amplitude at and below 1 kHz, with a roll-off of approximately 9 dB/octave.

We also determined the long-term average sound level (A-weighting) of road noise at 1m (mean ± S.D. = 68.0 ± 3.30 dB) and 20 meters (mean ± S.D. = 61.1 ± 3.04 dB) from the road edge with a Larson Davis Soundtrack LXT sound level meter. We used these LTAS measurements to determine the appropriate playback level to mimic a moderately busy roadway at our field sites. Finally, we generated 20 s of white noise in PRAAT with the same RMS amplitude as the chickadee and noise files, which we used to calibrate the sound levels in the field. We verified that these successfully produced the sound levels we desired for our noise prior to the start of our experiment.

We used an Anker SoundCore 2 speaker (frequency response: 70 Hz to 20 kHz; model 2A3105) connected to a DR-05 Tascam for playback of both the predator and anthropogenic noise stimuli. Prior to the start of our experiment, we checked the frequency response characteristics of our speaker in third-octave bands using a Larson Davis Soundtrack LXT sound level meter to verify that the frequency response was flat in the range of our stimuli. We also determined the sound field produced by our speakers. At a distance of 20 meters there was a loss of 8.5 ± 0.28 dB (mean ± S.D.) directly behind the speaker and a loss of 9.4 ± 0.95 dB (mean ± S.D.) 90° to the left or right of the speaker face. The loss directly above the speaker was (mean ± S.D.) 4.6 ± 0.56 dB. Thus, birds approaching the experimental area in front of the

speaker experienced higher noise levels than those approaching from the side or behind the speaker, although noise did travel in all directions. Sound fields are not typically reported for playback experiments and an uneven sound field is likely found in any playback experiment in which a single speaker is used. Therefore, an uneven sound field is not unique to our experiment, but we find it valuable to document and hope other authors will consider reporting this information as well. To verify that our stimuli were not distorted through the speaker, we recorded our stimuli after playback through the Tascam and speaker and cross-correlated our re-recorded stimuli with our original recordings in PRAAT.

Immediately prior to the start of each trial, we calibrated our playback speaker sound levels at least 100 m away from the experiment location to avoid disturbing the birds. We calibrated the amplitude of the *chick-a-dee* calls and anthropogenic noise to 75 dBA at 1 m with a Pyle PSPL05R sound level meter (fast integration time) using the previously described white noise calibration stimulus, in line with amplitudes we have recorded at our field sites. This resulted in a mean sound level of 72.9 ± 1.7 dB for the anthropogenic noise files when recorded at 1 m. This is slightly higher than the road noise levels we recorded at 1 m on a busy street. However, the sound levels at 20 m were lower (45–50 dBA) than those on busy a suburban street, but in line with sound levels during non-peak traffic (our single speaker noise source, rather than a continuous noise front consistent with passing traffic, resulted in lower amplitudes at a distance).

## Experimental setup

At each of the 14 locations we used a 1.7 m high feeding platform as the center of our experiment. A DR-05 tascam was attached to the platform to record vocal activity (settings in "call coding" below). We placed marking flags 5, 10, and 20 m from the platform in four directions (see **S1 Fig** for diagram of experiment set-up). For the anthropogenic noise playback, we placed the tripod 20 m from the platform in the direction of the nearest road. The speaker was placed on the tripod at a height of 1.4 m. For the predator presentation, we placed a tripod at a distance of 5 m from the feeding platform in the opposite direction of the nearest road. This tripod was set to a height of 1.7 m, as Northern saw-whet owls are known to hunt from this height [45] and we often observe chickadees vocalizing from this height. The tripod held either an Anker speaker (acoustic stimuli) or the taxidermy mount (visual stimuli). Both types of predator stimuli were initially covered by a wire framed camouflage fabric cylinder to diminish any response to novel stimuli during the pre-stimulus period. The fabric cylinder was positioned such that it could be lowered during the predator presentation period.

Before and after each trial, we used a Kestrel 3000 (Nielsen Kellerman USA) to measure mean wind speed (m/s), humidity (%) and temperature (˚C). We also recorded the background sound level using a Pyle PSPL05R sound level meter (fast integration, A-weighting) before and after the trial. Three observers were present for each trial and were positioned outside the 20 m radius to reduce bird response to human presence. One observer controlled anthropogenic noise playback, one observer controlled the predator reveal or the playback of *chick-a-dee* calls, and the primary observer was in a position that ensured visual coverage of the feeding platform. The three observers counted approaches of our target species within 5, 10, or 20 m of the feeding platform. An approach was recorded each time a bird crossed between and landed within any of the 5, 10, or 20 m boundaries around the feeding platform.

Each trial was 20 minutes in duration with a 5-minute pre-stimulus period, a 10-minute during-stimulus period during which the stimuli were played and/or revealed, and a 5-minute post-stimulus period. The predator stimulus was revealed by pulling a string to drop the camouflage cylinder and (in the case of the acoustic stimulus) begin playback (see S1 Fig for diagram).

The birds were exposed to the experimental stimuli for 10 minutes, after which the *chick-a-dee* call and anthropogenic noise playback ended. It was not possible to recover the owl in the field without disrupting the birds, so it remained uncovered during the post period. Therefore, in the main results below, we present two sets of data. First, we present the results for a dataset that includes only the pre and during periods. Second, we present the results from the dataset that included the pre, during, and post periods. Although the post period differs between the visual and acoustic stimuli, we believe the quiet period following the noise presentation is still of interest because altered behavior from predator signals can persist after the predator signals have ended. Additionally, while our playback of the call stopped after 10 minutes, calling of individuals in the area persisted in the post-stimulus time period. If masking was at play, the end of the noise playback might facilitate recruitment of additional individuals that are now able to detect the calling individuals. Therefore, we present both sets of data below.

## Call coding

We recorded vocalizations during the trials with a DR-05 Tascam as 16-bit mono wave files with a Tascam recording setting of 70 and a sampling rate of 44.1 kHz. We identified vocalizations of our target species from spectrograms generated in Raven Pro 1.6 (Hanning window size = 512 samples, 3dB filter bandwidth = 124 Hz). As anthropogenic noise playback could potentially prevent visual detection of vocalizations in the spectrograms (particularly lower frequency calls), we overlaid a recording of a type-matched anthropogenic noise file on the audio files of "quiet" experimental trials to minimize detection bias between files. The overlaid noise files were recorded at 20 m with the same recording equipment and settings used during the trial. Recordings were made at an area of the Vassar Farm and Ecological Preserve in which our target species were not present.

We counted the number of mobbing calls (*chick-a-dee* or *quank)* produced by each species during each of the three periods. We only included counts of birds that were likely to be responding to our stimuli by excluding calls with a signal to noise ratio less than that of a reference 65 dB *chick-a-dee* call recorded at 20 m. Finally, we counted the number of D notes in the chickadee and titmouse *chick-a-dee* calls, as well as the number of quank notes in each nuthatch *quank* call (Fig 1), as they have been previously shown to correlate with perceived predation risk for chickadees and titmice [37, 38] and are potentially related to anti-predator behavior [36] (but see [39]), or at least "excitement" [46].

## Statistics

We used R Studio (v4.2.0) to analyze all data using generalized linear mixed models [47]. Approaches, number of calls and number of notes were analyzed as the number per 10-minute interval. The number of notes in the pre and post period values were multiplied by 2 for consistency with the during period. We conducted four sets of analyses. First, we analyzed the number of approaches of our three target species. Second, we analyzed the number of each vocalization type produced by each species. Third, we analyzed the total number of targes notes produced, as well as the number of notes per call produced by chickadees (D notes) titmice (D notes) and nuthatches (quank notes). For all models, we first determined the distribution that best fit the data using the package fitdistrplus. A negative binomial distribution best fit the data in the approach models, call count models, and total number of notes models, while a gaussian distribution best fit the data in the number of notes per call models. To create the models, we used the glmer.nb function in the MASS package for negative binomial distributions and the lmer function in the lme4 package for gaussian distributions. Platform location was included as a random effect in all models, as our four trial types were repeated within

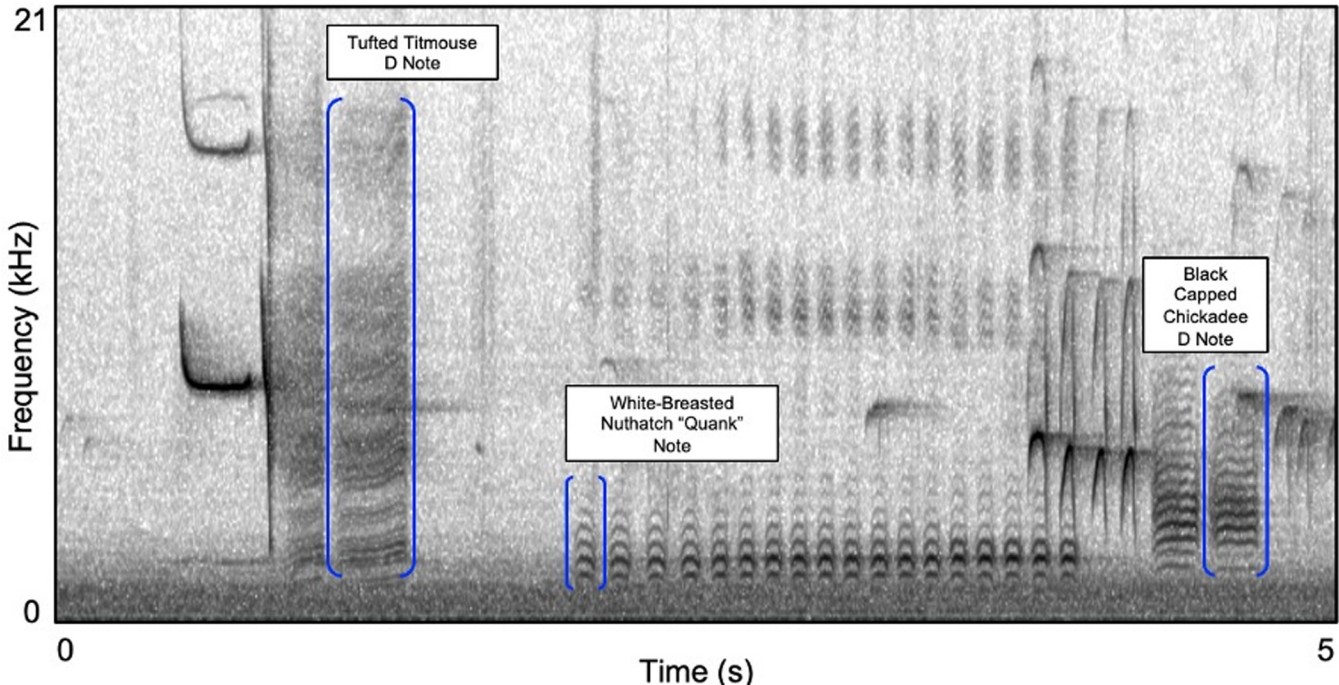

**Fig 1. Spectrographic representation of the alarm calls of our three focal species.** From left to right, brackets and labels indicate the *chick-a-dee* mobbing call of the tufted titmouse, *Baeolophus bicolor*, the *quank* alarm calls of the white-breasted nuthatch, *Sitta carolinensis*, and the *chick-a-dee* mobbing calls of the black-capped chickadee, *Poecile atricapillus*. (Hanning window, DFT = 512, 3dB filter bandwidth = 124 Hz).

each location. Additional random effects (e.g. humidity, cloud cover, etc.) did not sufficiently improve the models and were not included. Fixed effects and all their interactions were initially included in each model. Significant interaction effects and all lower order interactions it included were retained. If higher order interactions were not significant, we compared models with and without these interactions using the anova function in the stats package. If the model with that interaction produced a better model fit, that interaction was also retained. All other interactions were removed. Ultimately, we found that the full models and the reduced models were not qualitatively different; therefore, we present the full models here. After the final models were created, we used the Anova function in the car package to test for statistical significance.

The approach model included the fixed effects of predator stimulus modality (acoustic, visual), noise (presence, absence), period (pre, during), species (black-capped chickadee, tufted titmouse, white-breasted nuthatch), and their interactions. Note, that we did not analyze approaches at other distances because vegetation changes across time and between observer locations resulted in poor inter-observer reliability scores, reducing our confidence that accurate counts were obtained at these distances.

All nine of the vocalization models (three models for each species: one each for calls, total notes, and notes per call) included the fixed effects of predator stimulus modality (auditory, visual), noise (presence, absence), period (pre, during), and their interactions.

We only included trials in which the target species was present (i.e. if no nuthatches were present in a particular trial, we did not include that trial in any of the nuthatch call analyses).

We also ran each of the above models with the addition of the data for the post presentation period and include both sets of analyses in the results below. To allow readers to fully interpret the results, figures depict data from all three periods (pre, during, and post) when period was a

significant factor or part of a significant interaction in the model. For other main and interaction effects, we specify which periods were included in the figure captions. Also note that in one nuthatch trial a live predator appeared during the pre period, inflating the calling rate, and we therefore removed that trial from the models. All data are available in the electronic supplementary material (S1 Data).

# Results

## Approaches

In the model including only the pre- and during-periods, there was a significant main effect of species on the number of approaches within 0–5 meters of the feeding platform ($\chi^2_2 = 81.21$, $p < 0.001$, Fig 2A), with a greater number of chickadee approaches than nuthatch or titmice approaches. This is not surprising, as there are generally more chickadees present at our study sites than titmice or nuthatches. However, there were no significant main effects of period ($\chi^2_1 = 0.76$, $p = 0.383$), presence of noise ($\chi^2_1 < 0.01$, $p = 0.967$), or predator stimulus modality ($\chi^2_1 = 0.7$, $p = 0.70$). There was a significant species × predator interaction ($\chi^2_2 = 10.55$, $p = 0.005$, Fig 2B) whereby chickadees had slightly fewer approaches to the visual stimulus than the auditory stimulus. Titmice had significantly more approaches to the visual than the acoustic stimulus, suggesting that the visual stimulus may be a particularly salient cue for titmice. No other two-way interactions nor the three-way interaction were significant ($\chi^2 < 1.63$, $p > 0.188$). The results from the model including post period were qualitatively similar with a significant effect of species ($\chi^2_2 = 99.54$, $p < 0.001$), but no effects of period ($\chi^2_1 = 2.06$, $p = 0.357$), presence of noise ($\chi^2_1 = 1.50$, $p = 0.221$), predator stimulus modality ($\chi^2_1 = 0.86$, $p = 0.35$), nor any interactions ($\chi^2 < 1.63$, $p > 0.188$). As such, our analysis of approaches supports neither the distraction nor masking hypotheses.

## Number of calls

**Chickadees.** For chickadees in the model that included only the pre and during periods, the number of *chick-a-dee* calls was significantly affected only by the main effect of trial period (Table 1 and Fig 3A), with more calls produced during the predator stimuli presentation than produced in the pre period. All the presentations of predator stimuli increased the number of calls produced, supporting neither the masking nor distraction hypotheses. No other main or interaction effects were significant in this model (Table 1). This was quite different than the model that included all three periods, in which we found significant effects of predator stimulus type, noise, trial period and a noise × trial period interaction effect (Table 2). Chickadees produced more *chick-a-dee* calls when presented with the visual predator stimulus compared to the acoustic stimulus, which appeared to be driven primarily by a continued response to the predator mount during the post period, during which time the owl remained uncovered (Fig 3A); this was true in both noise conditions. Chickadees produced more *chick-a-dee* calls in the during and post period than in pre period (Fig 3A), suggesting that the stimuli effectively evoked responses in the chickadees. Finally, chickadees also produced more calls in quiet than in noise during the post-trial, suggesting the lingering effects of the stimulus may continue longer in a quiet environment (Fig 3B), which was true for both stimulus modalities, suggesting distraction may be at play when a longer time period is taken into account. We do note, however, that during the post period the owl presentation persisted without noise. However, this was true in both the noise and quiet trials, which we would expect to reduce the impact of noise, if anything. We did not find any other significant two or three-way interactions (Table 2).

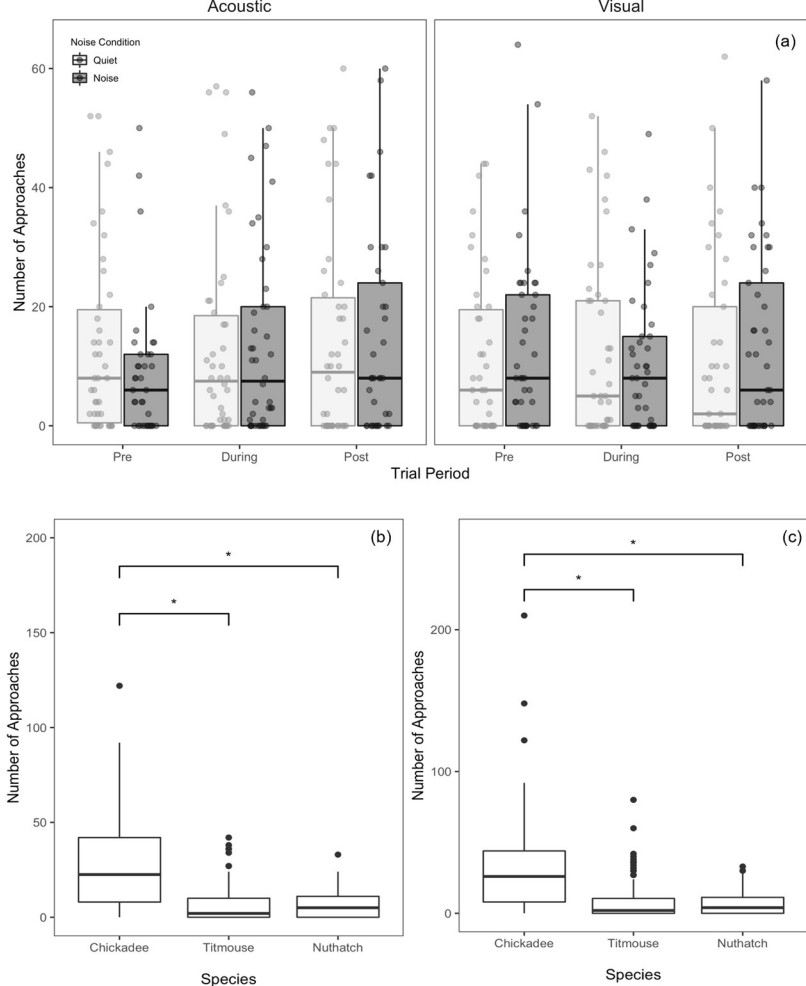

**Fig 2. Number of approaches within 5 meters of the feeding platform.** Data are presented as a function of (a) predator stimulus modality, noise presence, and period, (b) of species (including only the pre and during periods), and (c) of species (including the pre, during, and post periods). An approach was defined as a bird crossing a boundary and landing within 5m of the feeding platform. Asterisks indicate significant differences (p < 0.05) that were obtained using the anova function in R 1.4.1717. Boxes represent interquartile ranges, whiskers represent range, and central lines indicate median values. All data points are shown in (a), jittered along the x-axis for visibility, while only outliers are shown as points in (b) and (c).

**Titmice.** For titmice in both models, the number of *chick-a-dee* calls was significantly affected by predator stimulus modality and period, but not the presence of noise (Table 1). Titmice produced more *chick-a-dee* calls when presented with the visual predator mount compared to acoustic heterospecific calls in models without (Fig 4A) and with (Fig 4B) the post period included. Titmice also produced more *chick-a-dee* calls in the during period than in the pre period, with no other differences between periods (Fig 4C). We did not find any significant two or three-way interactions. Overall, the models both without and with the post period were qualitatively identical (Table 2) and suggest that titmice are largely unaffected by low levels of anthropogenic noise.

**Nuthatches.** For nuthatches, the number of *quank* calls was not significantly affected by any of the main effects in the models without (Table 1) or with (Table 2) post-stimulus period

**Table 1. Results of generalized linear mixed models that include only the pre and during-stimulus periods for our three species: Black-capped chickadees (N = 50 trials), tufted titmice (N = 39 trials), and white-breasted nuthatches (N = 44 trials).** For each species we separately analyzed the number of mobbing calls produced per 10-minute interval (binomial distribution), the number of notes produced per 10-minute interval (negative binomial distribution), and the number of notes per call (Gaussian distribution).

| Vocal Parameter | Fixed Effects | Black-capped Chickadee | | | Tufted Titmice | | | White-breasted Nuthatches | | |
|---|---|---|---|---|---|---|---|---|---|---|
| | | Chi-square | DF | P | Chi-square | DF | P | Chi-square | DF | P |
| Number of Calls | Predator | 0.24 | 1 | 0.63 | **4.24** | **1** | **0.040** | 0.35 | 1 | 0.553 |
| | Noise | 0.14 | 1 | 0.70 | <0.001 | 1 | 0.995 | 1.02 | 1 | 0.311 |
| | Period | **18.13** | **1** | **<0.001** | **13.24** | **1** | **<0.001** | 0.69 | 1 | 0.406 |
| | Predator × Noise | 0.54 | 1 | 0.461 | 1.45 | 1 | 0.229 | 0.77 | 1 | 0.379 |
| | Predator × Period | 1.70 | 1 | 0.19 | 0.01 | 1 | 0.933 | 1.20 | 1 | 0.273 |
| | Noise × Period | 0.05 | 1 | 0.82 | 0.55 | 1 | 0.457 | 0.01 | 1 | 0.923 |
| | Predator × Noise × Period | 1.54 | 1 | 0.21 | 0.07 | 1 | 0.787 | **5.35** | **1** | **0.021** |
| Number of Notes | Predator | 0.30 | 1 | 0.584 | **9.21** | **1** | **0.002** | 0.87 | 1 | 0.350 |
| | Noise | 0.73 | 1 | 0.394 | 0.05 | 1 | 0.819 | 3.20 | 1 | 0.073 |
| | Period | **13.54** | **1** | **<0.001** | **5.29** | **1** | **0.021** | 0.33 | 1 | 0.567 |
| | Predator × Noise | 0.26 | 1 | 0.611 | 1.84 | 1 | 0.174 | 0.47 | 1 | 0.494 |
| | Predator × Period | 2.29 | 1 | 0.130 | 0.48 | 1 | 0.487 | 2.35 | 1 | 0.125 |
| | Noise × Period | 0.005 | 1 | 0.945 | 0.96 | 1 | 0.327 | <0.001 | 1 | 0.999 |
| | Predator × Noise × Period | 0.68 | 1 | 0.409 | 1.37 | 1 | 0.242 | **6.41** | **1** | **0.011** |
| Notes Per Call | Predator | 2.47 | 1 | 0.116 | 1.22 | 1 | 0.269 | 1.94 | 1 | 0.164 |
| | Noise | 1.39 | 1 | 0.239 | 0.14 | 1 | 0.705 | 0.59 | 1 | 0.443 |
| | Period | 1.09 | 1 | 0.296 | 0.60 | 1 | 0.438 | 1.91 | 1 | 0.167 |
| | Predator × Noise | 2.85 | 1 | 0.092 | 0.27 | 1 | 0.605 | 0.78 | 1 | 0.377 |
| | Predator × Period | 0.40 | 1 | 0.530 | 0.05 | 1 | 0.832 | 1.16 | 1 | 0.281 |
| | Noise × Period | 2.04 | 1 | 0.153 | 0.02 | 1 | 0.900 | 0.003 | 1 | 0.953 |
| | Predator × Noise × Period | 1.75 | 1 | 0.186 | 0.11 | 1 | 0.745 | 0.18 | 1 | 0.67 |

Bold values indicate a statistically significant effect.

included. The three-way predator × noise × period interaction was significant in both models (Tables 1 and 2), while the two-way predator × noise interaction was only significant in the model with the post stimulus period included (Table 2). We did not find any other significant two-way interactions in either model. The driver of the three-way interaction in the model without post period included was an increase in calling from the pre to during-period in the quiet acoustic condition with little difference across other conditions (Fig 5). However, in the model with the post period included the three-way interaction also revealed differences in the post period (Fig 5). In general, the responses continued into the post period and strengthened in all the conditions except the noisy acoustic predator stimulus condition. In the post period the calling response to the acoustic stimulus was greater following quiet relative to noise. However, for the visual stimulus there was no difference in calling response in the periods following quiet and noise, suggesting a lingering effect of masking. Furthermore, there was no difference between the calling response to the visual and acoustic stimuli following the quiet presentations suggesting that in quiet both stimuli evoked calling behavior. However, the calling response to the visual stimulus following the noise presentation was greater than the response to the acoustic stimulus following the noise presentation. Together, these results suggest that masking may be the primary mechanism by which anti-predator responses are altered in anthropogenic noise for nuthatches.

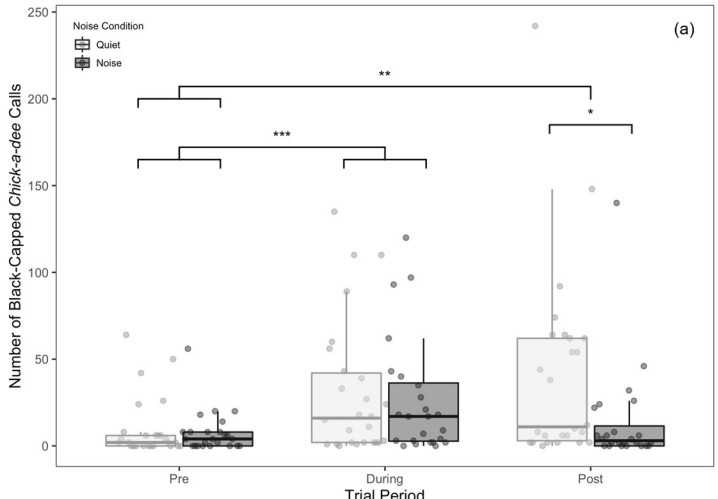
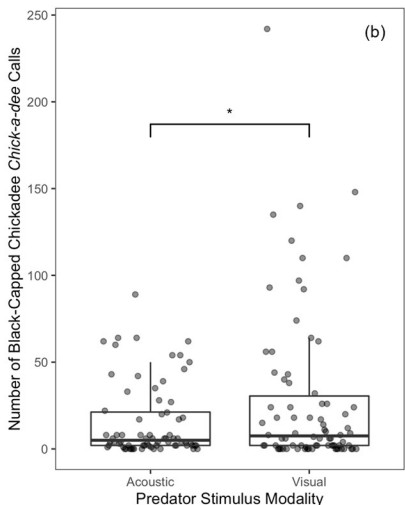

**Fig 3. Total number of black-capped chickadee *chick-a-dee* calls produced per 10-minute period.** Data are presented as a function of (a) period and noise presence and (b) predator stimulus modality (including pre, during and post periods). Calls of two or more *chick-a-dee* notes in a series were included. Asterisks indicate significant differences (* = p < 0.05) that were obtained using the anova function in R 1.4.1717. Boxes represent interquartile ranges, whiskers represent range, and central lines indicate median values. Individual data points are jittered along the x-axis for visibility.

## Total number of notes

**Chickadees.**   For chickadees in the model that included only the pre and during periods, the number of D notes produced was significantly affected only by the main effect of period (Table 1 and Fig 6A), with more notes produced during predator presentation period than in the pre-period (Fig 6A). No other main or interaction effects were significant (Table 1), suggesting that both stimulus modalities were effective, although neither masking nor distraction was apparent. This was quite different than the model that included the post period, in which the number of D notes produced by chickadees was significantly affected by the main effects of noise, period, and predator stimulus modality (Table 2). We also found a significant predator × period interaction effect. We did not find any other significant two or three-way interactions.

Chickadees produced more D notes in quiet than in noise (Fig 6B), an effect that persisted despite the continued presentation of the visual predator stimulus in the post period. This suggests that noise affected both stimulus modalities, indicating that distraction plays a role in changes in chickadee anti-predator behavior. Chickadees produced more D notes in the during and post periods than in the pre-period (Fig 6A), suggesting that both stimuli were effective. Chickadees produced more D notes when presented with the visual predator mount compared to acoustic conspecific calls (Fig 6A), which was driven by the during and post periods. This may be a result of the continued presence of the visual predator mount during the post period or greater certainty of predator presence. Overall, these results suggest that distraction is likely an important mechanism in chickadees.

**Titmice.**   For titmice in both models, the number of D notes was significantly affected by predator stimulus modality and period, but not noise or any of the interactions (Tables 1 and 2). Titmice produced more D notes when presented with the visual predator mount compared to acoustic heterospecific calls (Fig 7A and 7B). Titmice also produced more D notes in the during and post periods than in the pre-trial, with no other differences between periods (Fig 7C).

**Table 2. Results of generalized linear mixed models that included the pre, during, and post periods for our three species: Black-capped chickadees (N = 50 trials), tufted titmice (N = 39 trials), and white-breasted nuthatches (N = 44 trials).** For each species we separately analyzed the number of mobbing calls produced per 10-minute interval (binomial distribution), the number of notes produced per 10-minute interval (negative binomial distribution), and the number of notes per call (Gaussian distribution).

| Vocal Parameter | Fixed Effects | Black-capped Chickadee | | | Tufted Titmice | | | White-breasted Nuthatches | | |
|---|---|---|---|---|---|---|---|---|---|---|
| | | Chi-square | DF | P | Chi-square | DF | P | Chi-square | DF | P |
| Number of Calls | Predator | **3.86** | **1** | **0.049** | **4.65** | **1** | **0.031** | 0.05 | 1 | 0.824 |
| | Noise | **3.94** | **1** | **0.047** | 0.23 | 1 | 0.629 | 1.64 | 1 | 0.200 |
| | Period | **17.95** | **2** | **<0.001** | **15.88** | **2** | **<0.001** | 1.56 | 2 | 0.459 |
| | Predator × Noise | <0.001 | 1 | 0.97 | 0.006 | 1 | 0.94 | **5.33** | **1** | **0.021** |
| | Predator × Period | 3.22 | 2 | 0.20 | 1.56 | 2 | 0.459 | 3.75 | 2 | 0.153 |
| | Noise × Period | **6.31** | **2** | **0.043** | 2.12 | 2 | 0.346 | 0.08 | 2 | 0.961 |
| | Predator × Noise × Period | 1.72 | 2 | 0.42 | 4.89 | 2 | 0.087 | **6.94** | **2** | **0.031** |
| Number of Notes | Predator | **4.69** | **1** | **0.03** | **8.86** | **1** | **0.003** | 0.25 | 1 | 0.614 |
| | Noise | **4.99** | **1** | **0.025** | 0.10 | 1 | 0.748 | 3.30 | 1 | 0.069 |
| | Period | **15.91** | **2** | **<0.001** | **8.88** | **2** | **0.012** | 0.38 | 2 | 0.826 |
| | Predator × Noise | 0.31 | 1 | 0.58 | 1.08 | 1 | 0.299 | 3.26 | 1 | 0.071 |
| | Predator × Period | **6.77** | **2** | **0.034** | 1.68 | 2 | 0.431 | 4.39 | 2 | 0.112 |
| | Noise × Period | 2.72 | 2 | 0.26 | 1.70 | 2 | 0.427 | 0.42 | 2 | 0.812 |
| | Predator × Noise × Period | 0.73 | 2 | 0.69 | 2.19 | 2 | 0.334 | 5.18 | 2 | 0.075 |
| Notes Per Call | Predator | **5.29** | **1** | **0.022** | <0.001 | 1 | 0.989 | 0.35 | 1 | 0.556 |
| | Noise | **4.98** | **1** | **0.026** | 0.006 | 1 | 0.936 | 2.67 | 1 | 0.103 |
| | Period | 0.81 | 2 | 0.67 | 0.98 | 2 | 0.614 | 3.31 | 2 | 0.192 |
| | Predator × Noise | 2.09 | 1 | 0.15 | 1.11 | 1 | 0.292 | 0.11 | 1 | 0.736 |
| | Predator × Period | 0.56 | 2 | 0.76 | 2.65 | 2 | 0.266 | 4.57 | 2 | 0.102 |
| | Noise × Period | 3.08 | 2 | 0.21 | 0.53 | 2 | 0.769 | 1.33 | 2 | 0.513 |
| | Predator × Noise × Period | 1.79 | 2 | 0.41 | 0.15 | 2 | 0.927 | 1.26 | 2 | 0.533 |

Bold values indicate a statistically significant effect.

Together these results suggest that our stimuli were effective in evoking anti-predator responses and that these responses were affected by neither masking nor distraction.

**Nuthatches.** For nuthatches, the number of quank notes was significantly affected by the three-way predator × noise × period interaction in the model without post period included (Table 1 and Fig 8), while the three-way interaction was only marginally significant in the model with post period included (Table 2). Nuthatches increased the number of quank notes produced during the acoustic stimulus presentation in quiet, with no other post hoc differences (Fig 8), which suggests that nuthatches responded to conspecific mobbing vocalizations and that these responses were diminished in noise. Together with the call results, this suggests that masking may play a role in affecting anti-predator responses in nuthatches. No other significant effects were found in either model (Tables 1 and 2).

## Number of notes per call

The number of D notes per chickadee *chick-a-dee* call was not significantly affected by any model terms in the model without post period (Table 1). In the model with all periods included, the number of D notes per *chick-a-dee* call was significantly affected by predator stimulus modality and noise, but no other main effects or interaction effects in the model with post period included (Table 1). Chickadees produced more D notes per call to the visual predator stimulus than the acoustic stimulus (Fig 9A) and produced more D notes per call in quiet

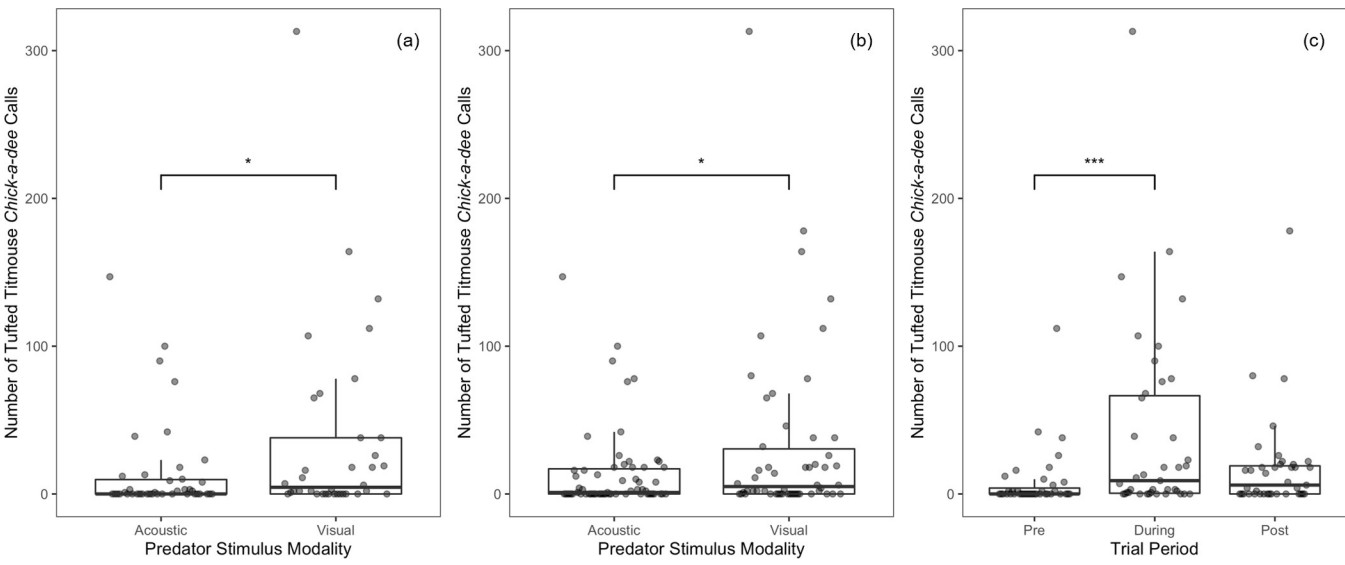

**Fig 4. Total number of tufted titmouse *chick-a-dee* calls produced per 10-minute interval.** Number of calls are presented as a function of (a) predator stimulus modality (for pre and during period only), (b) predator stimulus modality (for pre, during and post stimulus periods), and (c) trial period. Asterisks indicate significant differences (* = p < 0.05 | ** = p < 0.01 | *** = p < 0.001) that were obtained using the anova function in R 1.4.1717. Boxes represent interquartile ranges, whiskers represent range, and central lines indicate median values. Individual data points are jittered along the x-axis for visibility.

than in noise (Fig 9B). Calls with more D notes imply a greater perception of risk, thus these results suggests that distraction may be a mechanism that affects anti-predator vocalizations in chickadees.

The number of D notes per titmouse *chick-a-dee* call and the number of quank notes per nuthatch call were not significantly affected by any model terms in either model (Tables 1 and 2).

## Discussion

We found mixed support for the masking and distraction hypotheses, with different mechanisms affecting response to predators in each species. Our results for tufted titmice and white-breasted nuthatches were consistent across the models with and without the post period included. In tufted titmice we did not find support for masking nor did we find support for distraction, as there were no significant effects that included noise. However, titmice responded more strongly to the visual predator than the mobbing call playback in models including and excluding the post period, as is expected given the greater certainty of risk represented by a visual stimulus. For nuthatches, our data supported the masking hypothesis, as there was a three-way predator by noise by period interaction in models including and excluding the post period, suggesting that anthropogenic noise affected responses to mobbing calls more than it affected responses to visual predator stimuli in noise. However, the strongest effects of noise were time shifted to periods following the noise playback, with continuing or even increasing responses to the visual stimulus following both noise conditions, but continuing responses in the acoustic conditions only following the quiet presentations and not following the noise presentations. One possible explanation is that nuthatches integrated information from other species for a longer period of time than the ten minutes of our masking playbacks before upregulating their own calling. However, it is also important to note that while *quank* calls have been implicated in anti-predator behavior [35, 36], at least one study has shown that nuthatches did not produce more vocalizations to painted plastic hawk models

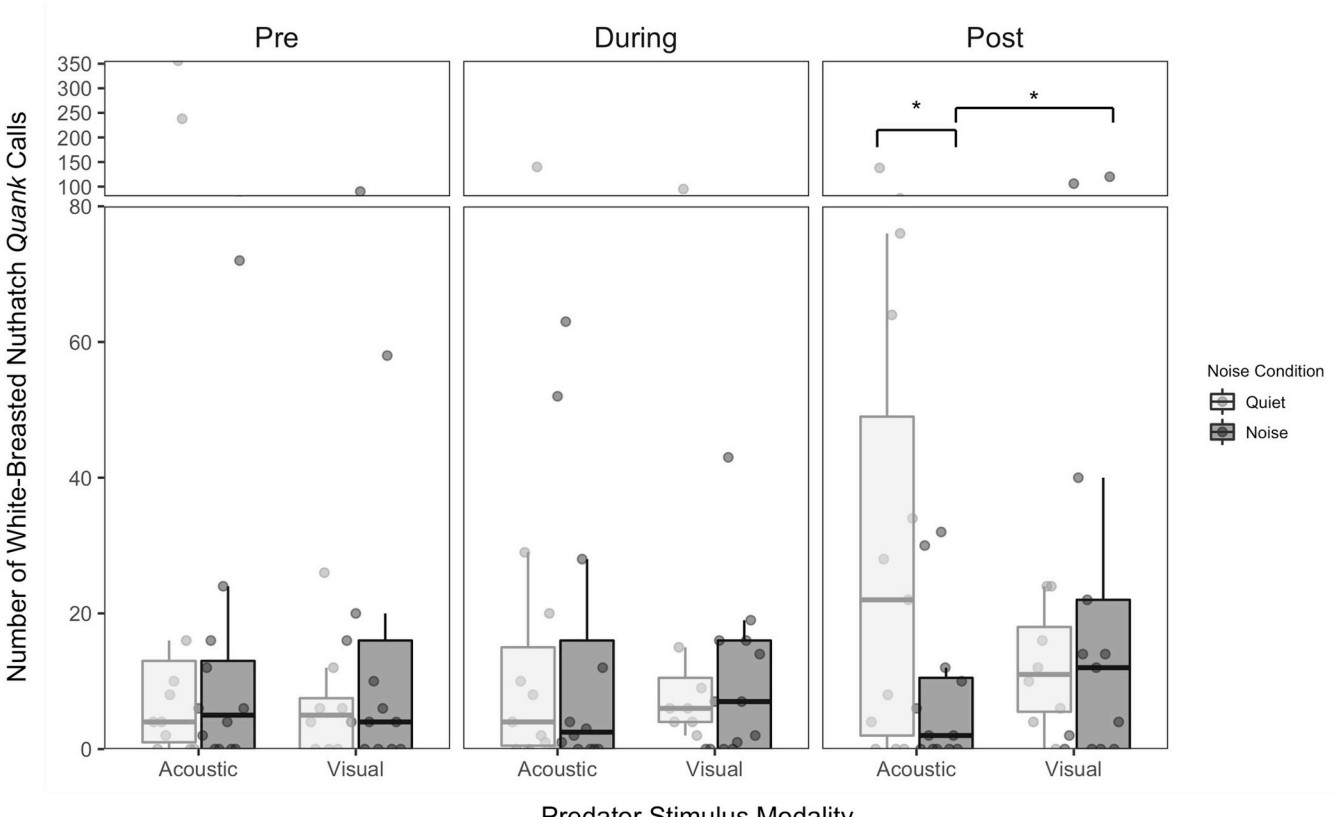

**Fig 5. Total number of white-breasted nuthatch *quank* calls produced per 10-minute interval.** Calls are presented as a function of predator stimulus modality, noise presence and period (all trial periods included). Boxes represent interquartile ranges, whiskers represent range, and central lines indicate median values. Individual data points are jittered along the x-axis for visibility. Asterisks indicate significant differences (* = p < 0.05) that were obtained using the anova function in R 1.4.1717.

than they did to similar dove models [39]. These results are largely consistent with previous findings on social roles and vocal production in all three of our study species.

The results for chickadees were more complicated. Period was a consistent predictor of the number of calls and the number of D notes produced by chickadees in models that both included and excluded the post period, suggesting that our acoustic and visual stimuli were effective at eliciting calling behavior. However, other patterns were different between the two sets of models. In particular, there was support for the distraction hypothesis in the chickadee models that include the post period, as we found an effect of noise, but no noise by predator stimulus modality interaction. This suggests that regardless of predator modality, noise can lead to reduced behavioral responses. However, in the model that excluded the post period, we did not find an effect of noise. This might suggest that the effects of distraction are relatively minimal in the low levels of noise used in our study and were not detectable without the power supplied by including the post period data. Note that the effect of continued presentation of the owl stimulus is equivalent in the post period for both the noise and no-noise conditions. If anything, we would expect the continued presentation of the owl to reduce the effect of noise as both were effectively "quiet" presentations of the predator. Previous work with this population found a robust decrease in chickadee anti-predator responses to titmouse *chick-a-dee* calls during playback of a higher amplitude anthropogenic noise stimulus [18]. This is consistent with the idea that the reduction in power from excluding post trial data may prevent us from identifying

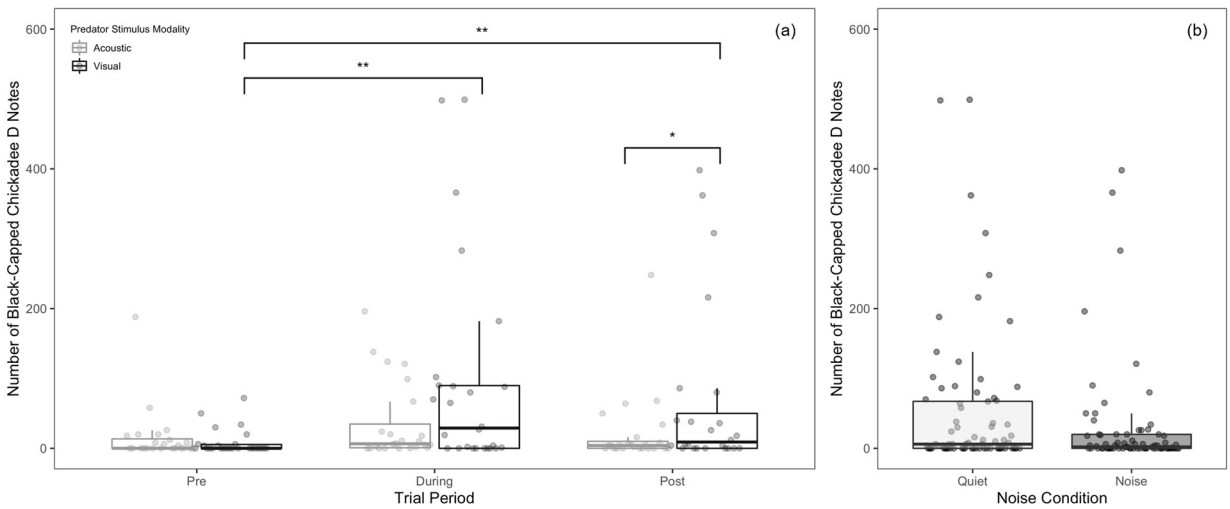

**Fig 6. Total number of black-capped chickadee D notes produced per 10-minute interval.** Note production is presented as a function of (a) predator stimulus modality and period and (b) noise presence (includes pre, during, and post periods). Asterisks for (a) indicate significant differences (* = $p < 0.05$ | ** = $p < 0.01$) that were obtained using the anova function in R 1.4.1717. Boxes represent interquartile ranges, whiskers represent range, and central lines indicate median values. Individual data points are jittered along the x-axis for visibility. Two outliers with counts above 600 under the visual condition in the post period were not included in the graphs for ease of interpretation but were included in the statistical analysis.

relatively weak changes in behavior in lower levels of noise. Additionally, we found that visual predator stimulus evoked a greater response than the acoustic predator stimuli in chickadees when we include the post period, but this effect was lost when the post period was removed. Stronger responses to visual predator stimuli are expected based on previous work [6].

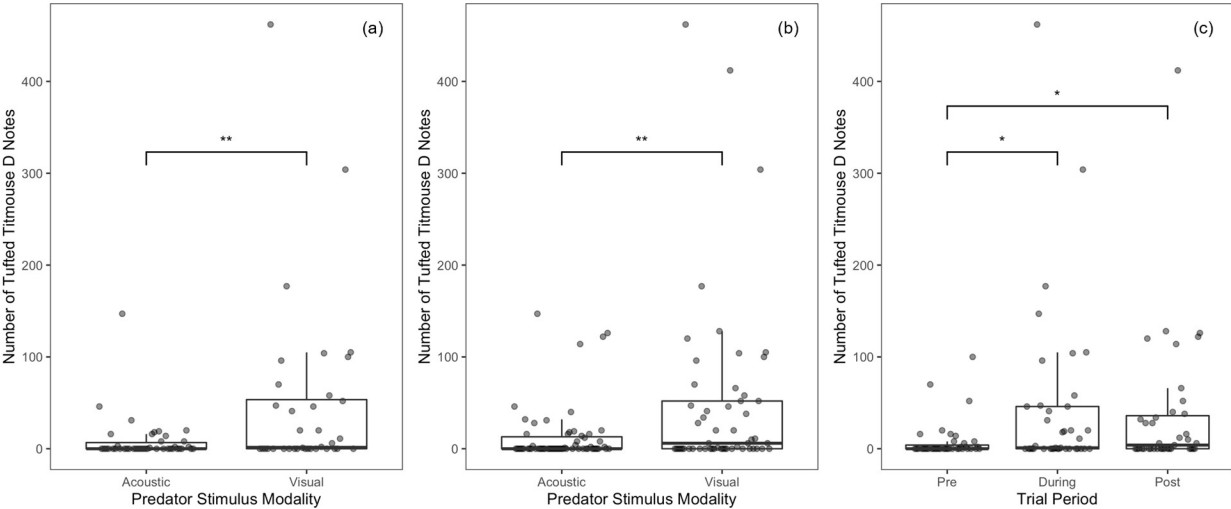

**Fig 7. Total number of tufted titmouse D-notes produced per 10-minute interval.** Note production is presented as a function of (a) predator stimulus modality (pre and during-trial only), (b) predator stimulus modality (including the pre, during, and post-period), and (c) period. Asterisks indicate significant differences (* = $p < 0.05$ | ** = $p < 0.01$) that were obtained using the anova function in R 1.4.1717. Boxes represent interquartile ranges, whiskers represent range, and central lines indicate median values. Individual data points are jittered along the x-axis for visibility.

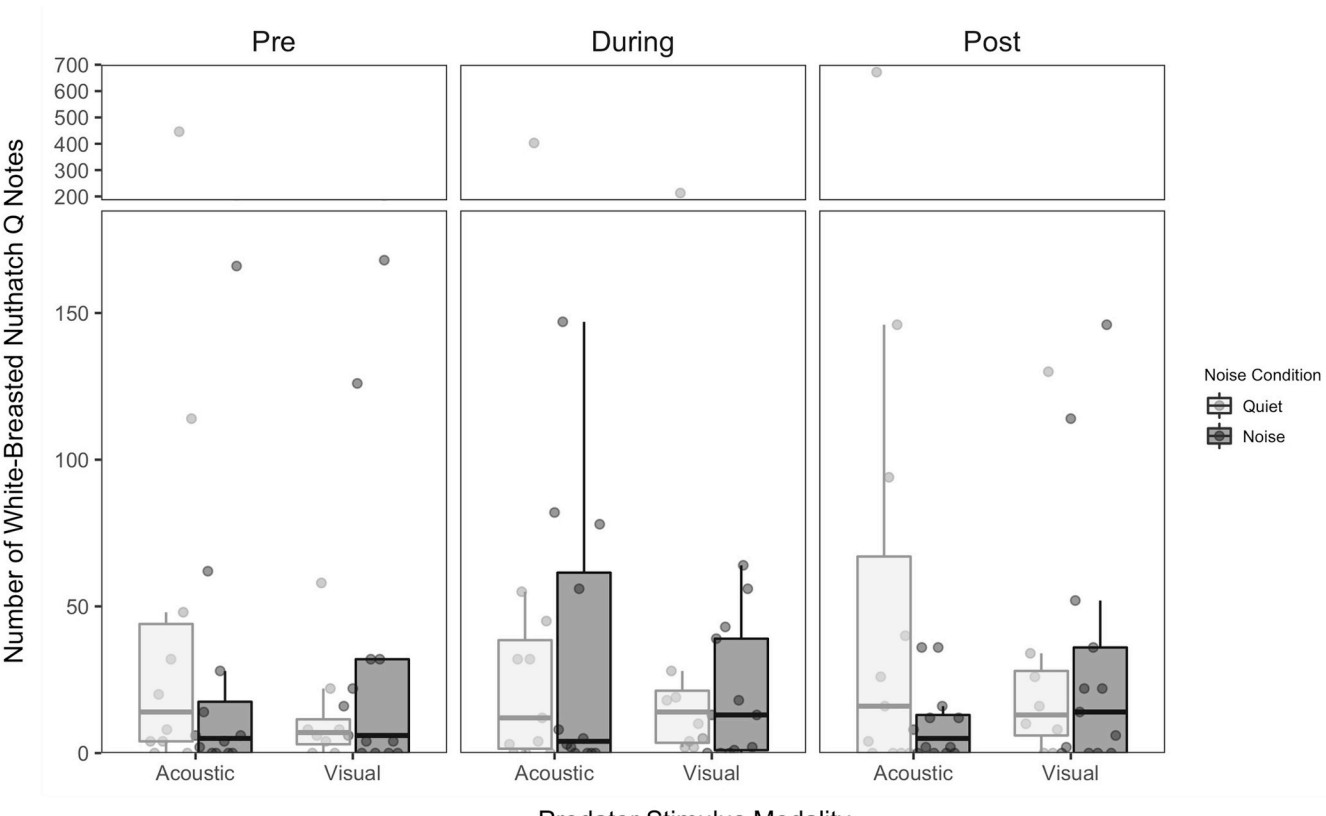

**Fig 8. Total number of white-breasted nuthatch quank notes produced per 10-minute interval.** Note production is presented as a function of predator stimulus modality and noise presence. The number of notes per 10 minutes. Boxes represent interquartile ranges, whiskers represent range, and central lines indicate median values.

The differences across species in the mechanisms regulating anti-predator signals may be a product of differences in the auditory processing of these species. The three species are known to differ in their frequency sensitivity, which is important in determining whether an animal can detect a low amplitude sound of a particular frequency [48–50]. Titmice have greater sensitivity at high frequencies than chickadees and nuthatches, which may enhance detection of high frequency alarm calls [48, 49]. Therefore, titmice may extract more information from the higher frequency components of calls, thereby escaping some of the masking effects at low frequencies in low amplitude noise.

Tufted titmice, white-breasted nuthatches and Carolina chickadees are also known to differ in their auditory filter bandwidth [51, 52] although auditory filter bandwidth has not been determined for black-capped chickadees. Auditory filters are a product of the mechanical and/or electrical tuning of the inner ear and determine the trade-off between frequency resolution and temporal resolution [50]. Animals with broader auditory filters must process signals with lower signal to noise ratios as more noise is processed with the signal of interest. Notably, while white-breasted nuthatches and tufted titmice did not differ significantly in filter bandwidth, but they did differ in K', a parameter that describes the efficiency of signal extraction in noise [51]. Titmice are more efficient (i.e. experience less masking) at extracting signals from noise than nuthatches (i.e. experience more masking). Therefore, at low levels of noise nuthatches may be more susceptible to masking than titmice, consistent with our results here. Carolina chickadees have broader filters than titmice and nuthatches at 2 kHz and similar filter

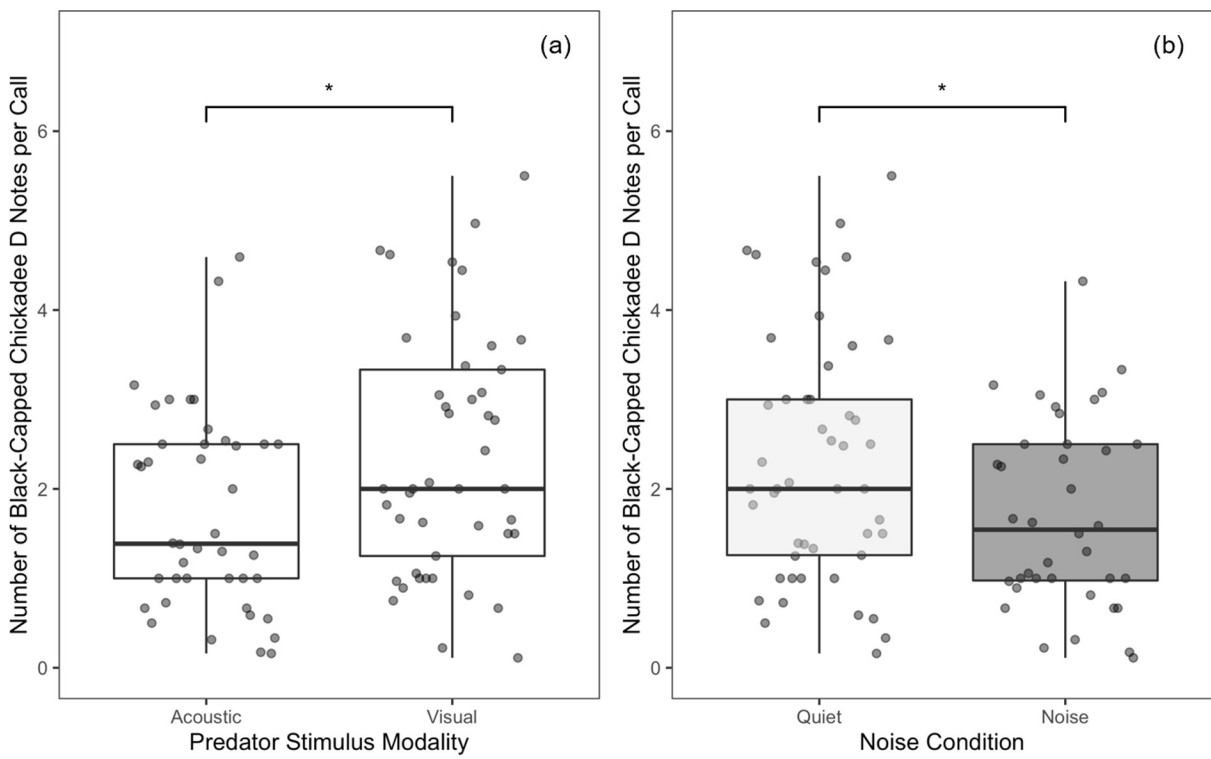

**Fig 9. Number of black-capped chickadee D notes per *chick-a-dee* call.** Data are collapsed over the pre, during and post periods and presented as a function of (a) predator stimulus modality and (b) noise presence. Asterisks indicate significant differences (* = p < 0.05) that were obtained using the anova function in R 1.4.1717. Boxes represent interquartile ranges, whiskers represent range, and central lines indicate median values.

widths at 4kHz. Moreover, Carolina chickadees have a K' that lies in between that of titmice and nuthatches, suggesting an intermediate ability to extract signals from noise [52, 53]. If black-capped chickadees are similar to Carolina chickadees in their filter configuration, they may be more susceptible to masking than titmice but less susceptible than nuthatches. This could also explain the inconsistent results for chickadees in the two models, with additional power in the models that included post-period allowing us to detect relatively small effects of noise. Obviously, this is speculative at this time, although recent work in our lab on auditory processing seem to recapitulate these patterns in tufted titmice, white-breasted nuthatches, and black-capped chickadees. We have also found that black-capped and Carolina chickadees are similar in other aspects of auditory processing, such as frequency sensitivity [49]. Although we found some support for distraction in chickadees, this is not mutually exclusive to masking, which may also contribute to reduced responses to call playbacks in noise.

Distraction is likely responsible for reduced anti-predator response in a broad array of taxa [13, 54–56]. In great tits, an anthropogenic noise proxy was found to increase foraging latencies for visually cryptic targets, but not for more obvious targets [55]. This suggests that noise competes for attentional space within challenging visual processing tasks (e.g. cryptic prey detection), however, when less attention is necessary (e.g. conspicuous prey detection) noise ceases to be a distraction [55]. In zebra finches, anthropogenic noise has been shown to decrease non-auditory cognitive performance, which suggests noise may be distracting or increases cognitive load [57]. Furthermore, there is evidence for distraction in hermit crabs, with louder white-noise causing longer hide latencies [13] and lab rats, with background noise

decreasing performance in operant-conditioned tasks [54]. Our results suggest that distraction may be the primary mechanism regulating anti-predator responses of chickadees in low levels of noise; although the effect size of this distraction may be relatively low. However, the effects of noise across modalities does not preclude masking from also influencing behavioral responses. Anthropogenic noise decreased the perception of alarm calls in great tits in a manner consistent with masking [19]. Great tits were unable to frequency shift their calls to restore the active space of their alarm signals.

Much of the work on the effects of anthropogenic noise on songbirds has focused on vocal production and proposed masking as a mechanism to explain changes. Some songbirds can pitch-shift their songs to higher frequencies in anthropogenic noise, which suggests that they may be trying to avoid auditory masking by increasing frequency [27]. This upward pitch shift may also be an acoustic byproduct of increasing the amplitude of their songs [58]. Much subsequent work has also found small pitch shifts in song in the presence of anthropogenic noise [28, 31, 59–62]. More recently, attention has turned to the effects of masking noise on anti-predator communication behaviors. For instance, anthropogenic noise reduced chickadee approaches to titmouse mobbing calls [18]. Additionally, nuthatches and titmice decreased their response to predator vocalizations with increasing anthropogenic noise, suggesting that the predator cue was masked and therefore going undetected [63].

While most work has focused on either distraction or masking in animal behavior literature, our evidence suggests that the mechanism producing reduced anti-predator communication may vary between species. We suspect that there is also a relationship between the mechanism at play and the amplitude of the noise. In environments where the levels of anthropogenic noise are high (e.g. near busy freeways or airports), masking may be the predominant mechanism across species, because the spectral energy in the noise is more likely to produce signal-to-noise ratios that are lower than those needed to detect mobbing and/or alarm calls. Contrastingly, in environments with lower levels of noise, the spectral overlap with calls may not be sufficient to effectively, or at least fully, mask calls. Contrary to this assumption, our results show that, even in relatively low levels of noise, responses to predation cues are diminished in some species. In these cases, masking, distraction, or some combination may still be at play. The relative contribution of each mechanism may depend on the auditory system of the species in question, as well as their receiver psychology and the behavioral or ecological roles they play in mixed species flocks.

Our evidence suggests that anti-predator responses are regulated by masking in nuthatches and to some extent by distraction in chickadees, which we speculate could be related to species-specific behaviors that reflect their social roles in our mixed-species flocks. Chickadees are known to be "sentinel" or "focal" species, gathering personal information about the presence of predators and producing temporally-precise alarm calls that alert conspecifics and heterospecifics of risk [64–66]. The sentinel behavior may incur a high cognitive load, as individuals are regularly acquiring, processing, and responding to information about predation risk. Noise has been found to impair cognitive function [57] and therefore, competing stimuli, regardless of the sensory domain, may reduce the ability of chickadees to acquire and respond appropriately to predation cues. On the other hand, nuthatches typically occupy a satellite position in flocks, primarily acquiring public information about predation risk from heterospecific alarm calls through "eavesdropping" [67]. This position in the flock is likely a result of their foraging behavior (i.e., clinging closely to tree trunks) and resulting morphology, which limits the visual coverage nuthatches have of their environment [35, 46]. Detecting conspecific calls may be a less challenging cognitive task, as these calls have easily detectable and localizable structures. However, we suggest that efficient call detection may be particularly sensitive to masking, especially if nuthatches are sensitive to the structure or rate of calling. Titmice are known to be

more bold than other species in their mobbing responses, getting closer to and remaining near predators for a longer time [68, 69]. A greater predisposition to mob, combined with efficient extraction of signals from noise [51], may explain the lack of an effect of noise on the vocal behavior of tufted titmice. Other work on mobbing behavior in noise supports this theory; it has been found that song sparrows do not react differently to temporally-consistent versus temporally-fluctuating noise, suggesting that they are not exploiting un-masked gaps via dip listening (Sweet et al., 2022). Instead, they increased vigilance in response to overall noise levels as though perceiving overall increased risk, which may be a byproduct of distraction [70]. Additionally, black-capped chickadees produce introductory notes and D notes with lower peak frequencies in noise, which is opposite to what we would expect if they were pitch shifting to avoid being masked, given that anthropogenic noise occupies lower frequencies [27, 62]. The maximum frequency of D notes has also been shown to decrease when D notes are produced in response to a predator [71] suggesting that lower notes might convey greater threat perception.

Finally, in nuthatches the effects of noise appeared to persist into the post period for acoustic stimuli. Responses to the visual predator remained elevated in both noise conditions, while responses to the acoustic predator stimulus were suppressed following the noise playback, but remained elevated following the quiet playback. Nuthatches also produced the lowest frequency alarm calls, lacking the high-frequency introductory notes of chickadees and titmice. Therefore, nuthatch *quank* calls may fail to recruit other nuthatches during the playback of the acoustic stimulus in noise, leading to a persistent decrease in the post period. Alternatively, the time-shifted decrease in alarm calling may be due to environmental awareness; nuthatchesmay not produce calls when they know they won't be heard and therefore fewer individuals may be recruited to mob.

Our study contributes to a growing number of papers that highlight previously unappreciated complexities in anthropogenic noise's effects on predation response. Asymmetries in mobbing behavior and specialized roles in social networks may influence how certain species are able to cope with increasing anthropogenic noise. Additionally, there has been surprisingly little work on how the auditory system of songbirds copes with both temporally-stable and fluctuating noise, which would inform species difference in the susceptibility to noise, particularly lower amplitude urban and suburban noise. Future comparative studies of masked thresholds and/or critical ratios will be very valuable in explaining the responses of different species to noise.

## Supporting information

**S1 Fig. Experimental setup. (A)** Each trial location consisted of a feeding platform, an anthropogenic noise speaker 20 m from the platform in the direction of the nearest road, and a predator mount holding the owl or chickadee call speaker 5 m away from the platform in the opposite direction. Flags were placed around the platform at 5 m, 10 m, and 20 m away from the platform. One observer was located near the anthropogenic noise speaker, another behind the predator mount, and a third observer stood in the direction that gave them clearest view of the experimental area. All three observers were stationed outside the experimental radius. **(B)** The predator reveal mechanism was operated by an observer standing 20 m away, who pulled a string to remove supports beneath the camouflage cylinder, which would cause the cylinder to fall and reveal the predator stimulus.
(TIF)

**S1 Data. Spreadsheet containing the data and metadata for the experiment.**
(XLSX)

## Acknowledgments

Thanks to Glenn Proudfoot and Miriam Cubstead for their assistance with producing our taxidermy model.

## Author Contributions

**Conceptualization:** Trina L. Chou, Anjali Krishna, Julia Goldberg, Megan D. Gall.

**Data curation:** Trina L. Chou, Anjali Krishna, Mark Fossesca, Avani Desai, Megan D. Gall.

**Formal analysis:** Trina L. Chou, Benjamin M. Basile, Megan D. Gall.

**Funding acquisition:** Megan D. Gall.

**Investigation:** Trina L. Chou, Anjali Krishna, Mark Fossesca, Avani Desai, Julia Goldberg, Sophie Jones, Benjamin M. Basile, Megan D. Gall.

**Methodology:** Trina L. Chou, Anjali Krishna, Mark Fossesca, Avani Desai, Julia Goldberg, Benjamin M. Basile, Megan D. Gall.

**Project administration:** Megan D. Gall.

**Supervision:** Megan D. Gall.

**Visualization:** Trina L. Chou, Anjali Krishna.

**Writing – original draft:** Trina L. Chou, Anjali Krishna, Mark Fossesca, Avani Desai, Morgan Stephens, Megan D. Gall.

**Writing – review & editing:** Trina L. Chou, Anjali Krishna, Mark Fossesca, Avani Desai, Julia Goldberg, Sophie Jones, Morgan Stephens, Benjamin M. Basile, Megan D. Gall.

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
