## [Decision Letter · Decision Letter 0]

5 Jun 2023

PONE-D-23-10182Mixed support for the effects of masking and distraction on anti-predator behavior in suburban anthropogenic noisePLOS ONE

Dear Dr. Gall,

Thank you for submitting your manuscript to PLOS ONE. After careful consideration, we feel that it has merit but does not fully meet PLOS ONE’s publication criteria as it currently stands. Therefore, we invite you to submit a revised version of the manuscript that addresses the points raised during the review process.

Both reviewers appreciated the work, but it would be important to provide missing details and justifications, and reassess the results on this basis as indicated by the reviewers.

We look forward to receiving your revised manuscript.

Kind regards,

Vivek Nityananda

Academic Editor

PLOS ONE

Journal Requirements:

4.Thank you for stating the following in the Acknowledgments Section of your manuscript: 

"Special thanks to the Vassar Undergraduate Research Summer Institute, the Vassar College Biology Department, and the Asprey Center for Collaborative Approaches to Science for contributing funding to this project. Thanks also to Glenn Proudfoot and Miriam Cubstead for their assistance with producing our taxidermy model."

5. We note that Figure ESM 2 in your submission contain copyrighted images. All PLOS content is published under the Creative Commons Attribution License (CC BY 4.0), which means that the manuscript, images, and Supporting Information files will be freely available online, and any third party is permitted to access, download, copy, distribute, and use these materials in any way, even commercially, with proper attribution. For more information, see our copyright guidelines: http://journals.plos.org/plosone/s/licenses-and-copyright.

a. You may seek permission from the original copyright holder of Figure ESM 2 to publish the content specifically under the CC BY 4.0 license. 

Reviewers' comments:

Reviewer's Responses to Questions

**Comments to the Author**

1. Is the manuscript technically sound, and do the data support the conclusions?

Reviewer #1: Yes

Reviewer #2: Yes

2. Has the statistical analysis been performed appropriately and rigorously? 

Reviewer #1: Yes

Reviewer #2: Yes

3. Have the authors made all data underlying the findings in their manuscript fully available?

Reviewer #1: Yes

Reviewer #2: Yes

4. Is the manuscript presented in an intelligible fashion and written in standard English?

Reviewer #1: Yes

Reviewer #2: Yes

5. Review Comments to the Author

Reviewer #1: I have read the manuscript entitled “Mixed support for the effects of masking and distraction on anti-predator behavior in suburban anthropogenic noise”. Overall, the authors found a simple and novel way to distinguish between masking and distraction in a field setting for three focal bird species. This is commendable! I believe this study will make a significant contribution to our understanding of how anthropogenic noise can lead to masking and distraction in several bird species. By streamlining and reorganizing the presentation of their findings, this manuscript will be much more accessible and will better emphasize the novelty and importance of their work. In its current state, the main message(s) are lost in the details but with extensive revisions, it will reach its full potential.

Title: It may be worth iterating that what this study found is interspecific differences, which is much more compelling!

Introduction:

Overall, the introduction was pleasant to read by avoiding unnecessary jargon and ensuring a great flow within and among paragraphs.

Lines 108–120: Using the question format interrupts the flow of the paragraph and the topic sentence is not very strong. This paragraph should not only outline purposes but also outline the scope of your study and state the value of your research. Revise to include more concrete details pertaining to your study.

Methods:

Specifics related to when and where the study was done, including coordinates, would be best presented at the beginning of the methods section to help situate the reader and as is common practice.

Lines 123–133: Revise as goals should be emphasized in the introduction, while your methods should focus on what was done and how. Consider using a more concise way of relating methods to goals: To determine the effects of X on Y, we did Z. Additionally, predictions should be in the introduction and not the methods section (i.e., 127–129).

Overall, the methods are scientifically sound and creative! Consider streamlining the methods section to focus on key aspects of the experimental setup and analysis, and relocate the more specific details to the supplementary materials.

Results: Consider re-organizing and focusing on key findings. I would suggest re-evaluating the necessity of all included figures, and consider moving some of them, along with certain methodological details, to the supplementary materials. This could include data that supports but does not directly contribute to the main narrative of your manuscript.

Discussion: I believe that after reviewing the manuscript to emphasize important findings, this section will have a clearer narrative. In its current state, it is difficult to relate to directly relate the discussion to the research questions and to the broader literature.

Citing: Throughout the manuscript, the authors indicate whether references were the topic was reviewed. In these instances, I suggest they opt for a more concise way of referencing these as to avoid continuously interrupting the flow of the text. It could be replaced by “see” as it is common practice. Additionally, instead of including a long list of citations, particularly in the introduction, consider only using a few representative citations as examples: (e.g., REF1, REF2, REF3).

Reviewer #2: The manuscript titled ‘Mixed support for the effects of masking and distraction on anti-predator behavior in suburban anthropogenic noise’ is a well written with proper introduction and discussion. However, the methods part is not up to the mark. There are many loop holes which makes it hard to follow. I have listed my comments as follows

Major comments

Line 112-114: ‘we presented mixed-species……presence (playback of chickadee chick-a-dee mobbing calls)’. Authors have not mentioned the rationale behind using chickadee mobbing calls as acoustic cues? Do all three species always occur together? I also wonder why the call of predator (Norther saw-whet owl) was not used as acoustic cues. A general background of relation between Chickadee, Tuftmice and Nuthatch is missing in the introduction part.

I suggest authors to add justification on using chick-a-dee call of Chickadee and also add a brief about all three species.

In Line no 153-154: It is mention that Northern saw-whet owls are viewed as high threat predators by black-capped chickadees and Chickadees produce vocalizations that encode a high degree of threat to saw-whet owls. However, in line 166-167 it is mentioned that 'The acoustic cue of predator presence consist of chickadee chick-a-dee calls with 3-4D notes that were arrange in 22 or 23 bouts'. Further, in line 172 -173, it states 'Calls of this type have previously been shown to communicate a moderate level of threat’. Thus, statements are confusing and therefore I feel that the chick-a-dee call used as stimulus is not appropriate. It is not clear why authors used a moderate level threat call instead of high-level threat. Justification needed.

Line 206-211: Authors have mentioned that they have calibrated the amplitude of chick-a-dee calls and anthropogenic noise to 75 dBA at 1 m. However, the natural amplitude of chick-a-dee call is not mentioned. I suggest authors to add the natural amplitude of chick-a-dee call of chickadee.

Line 223-224: ‘This tripod was set to a……this height’, since the study focus on three species it would be better if authors could add the foraging height of other two species as well.

Line 266: Authors claimed that they have counted number of songs and mobbing calls, however, same is not mentioned in result part.

Line 267: Authors also claimed that they have counted number of gargles made by chickadees and the number of high zees made by titmice. Though they have also mentioned that statistical analyses was not performed, I suggest that it will be better to add the result of this part as well. In addition, it will give reader more sense of the study if authors could provide information on what gargles and zees are.

Line 272: ‘Finally, we counted……………………. chick-a-dee calls’, this statement is confusing as authors haven’t mentioned before that both chickadee and titmouse both produce chick-a-dee call. One would misunderstand that chick-a-dee is only produce by chickadee. Thus, to avoid confusion, I suggest to add briefly about the calls of three species.

Line 272-274: Authors haven’t mentioned what they speculating by counting the number of each note types in chickadee and titmouse chick-a-dee calls and the number of quank notes in nuthatch quank call. I suggest authors to add what they are speculating by counting number of each note types and why it is important. Furthermore, according to Bartmess-LeVasseur et al.2010, no effect predator context in associated with quank call of Nutchatch. So how do you justify this statement?

Bartmess-LeVasseur, J., Branch, C. L., Browning, S. A., Owens, J. L., & Freeberg, T. M. (2010). Predator stimuli and calling behavior of Carolina chickadees (Poecile carolinensis), tufted titmice (Baeolophus bicolor), and white-breasted nuthatches (Sitta carolinensis). Behavioral Ecology and Sociobiology, 64, 1187-1198.

Line 284: Figure 1 do not give clear vision of different notes of chick-a-dee call produce by bot chickadee and titmouse. I suggest authors to add a better spectrogram showing different note types of all three species.

Line 293: ‘First, we ………………… species’, Since, experiment was conducted in wild habitat, how authors have controlled the abundance of each species during the experiment? Unequal population size of three species will lead to false estimation of the number approaches of three target species. Therefor, I suggest authors to provide information about the same.

Line 295-296: ‘we analyzed …………………….. and titmice’. Authors haven’t mentioned why they analysed D note per trial period, why not other notes. As the large audience do not have the background of chick-a-dee calls, it is not easy to follow the statement. I suggest authors to add the importance of D note in chick-a-dee call and how change in number of D notes changes the information content of the call.

Line 314-315: We analyzed ..……... one model. Though authors have normalised the uneven distribution of time which is mentioned in Supporting Information ESM2. I suggest authors to mentioned the same in main text. Same goes to line 325 -326.

Line 314-315: Authors have mentioned that they have normalised the time period by multiplying by 2 (also mentioned in Supporting Information ESM2) in the case of pre and post-trials. However, I speculate that number of approaches should decrease by time, for instance, if total number of individual present is 10 and 7 individual approaches in 5 minutes then number of individuals approached in next 5 minutes will be 3 not 7. Therefore, I suggest authors to instead of analysing for 10 minutes, analysing approach for 5 minutes will give better representation of data.

Line 548: ‘Therefore, titmice ………….. amplitude noise’. To support this statement, authors should provide the frequency range of noise which will help in understanding the range of frequency masking. If the frequency overlap between noise and chick-a-dee call is <2 kHz, then I suspect that Titmice responding is not just because of greater sensitivity at high frequencies as sensitivity range of chickadee is 2-4 kHz (Wong & Gall 2015).

Line 52: “masking, distraction, anthropogenic noise, anti-predator behavior”, These words are already present in the title. Key words should be different from the words present in the title.

Line 79: “(Damsky & Gall, 2017, Templeton et al. 2016)”, put reference in chronological order

Line 90: “Dooling and Blumenrath” change ‘and’ to ‘&’

Line 116: “anthropogenic traffic noise”, there is no natural traffic noise. It can be change to traffic noise.

Line 138: “Platforms were at least 0.4 km….”, it is not clear what authors refer platform to. Is it equivalent to territory size? If so, what is the territory range of these species?

Line 222: ‘we placed a tripod 5m’, may change to ‘we placed a tripod at a distance of 5 m’

Line 486: ‘Number of Notes Per Call’ may replace it with ‘Number of D Notes Per Call’

Line 536: ‘(Damsky and Gall, 2017)’ change it to (Damsky & Gall, 2017)

Line 546 & 555: ‘(Henry et al. 2017)’ change it to ‘(Henry et al., 2017)’

Line 552: ‘(Henry & Lucas 2010, Henry & Lucas 2011)’ add ‘,’ after ‘Lucas’

Line 565: ‘(Henry et al. 2011)’ add ‘,’ after ‘al.’

Line 577 &585: ‘Chan et al. 2010’ add ‘,’ after ‘al.’

Line 649: ‘(Sweet et al. 2021)’ add ‘,’ after ‘al.’

6. PLOS authors have the option to publish the peer review history of their article (what does this mean?). If published, this will include your full peer review and any attached files.

Reviewer #1: No

Reviewer #2: No

---

## [Author Response · Author response to Decision Letter 0]

14 Jun 2023

Response to reviewers

Reviewer #1: I have read the manuscript entitled “Mixed support for the effects of masking and distraction on anti-predator behavior in suburban anthropogenic noise”. Overall, the authors found a simple and novel way to distinguish between masking and distraction in a field setting for three focal bird species. This is commendable! I believe this study will make a significant contribution to our understanding of how anthropogenic noise can lead to masking and distraction in several bird species. By streamlining and reorganizing the presentation of their findings, this manuscript will be much more accessible and will better emphasize the novelty and importance of their work. In its current state, the main message(s) are lost in the details but with extensive revisions, it will reach its full potential.

Response: Thank you for your comments on our manuscript. We have made changes to the introduction and methods as suggested. We ultimately felt that it was best to leave the results as they stand to make it easier for readers to compare between the two models. However, we have added a few statements to draw attention to some of the main messages of the result. We are open to additional changes if the reviewer has specific items they would like us to consider. 

Title: It may be worth iterating that what this study found is interspecific differences, which is much more compelling!

Response: Thanks for the suggestion. We have updated the title. It now reads: 

Interspecific differences in the effects of masking and distraction on anti-predator behavior in suburban anthropogenic noise

Introduction:

Overall, the introduction was pleasant to read by avoiding unnecessary jargon and ensuring a great flow within and among paragraphs.

Lines 108–120: Using the question format interrupts the flow of the paragraph and the topic sentence is not very strong. This paragraph should not only outline purposes but also outline the scope of your study and state the value of your research. Revise to include more concrete details pertaining to your study.

Response: We have revised this to remove the question format and have added some of the material from the first paragraph of the methods. We hope this addresses the reviewer’s concern. For this comment and the next two comments, the relevant section now reads: 

Here, we investigated (1) whether temporally-fluctuating suburban noise is sufficient to alter the behavioral responses of black-capped chickadees (Poecile atricapillus, hereafter chickadees), tufted titmice (Baeolophus bicolor, hereafter titmice), and white-breasted nuthatches (Sitta carolinensis, hereafter nuthatches) to predator cues and (2) the relative contributions of the two proposed mechanisms, masking and distraction, to the effect of anthropogenic noise on anti-predator behavior. These three species form mixed species flocks during the non-breeding season, engage in predator mobbing together, and are known to respond to the mobbing calls of chickadees [34-35]. Chickadees and titmice both produce chick-a-dee calls and nuthatches produce quank calls in anti-predator contexts [35-37]. The chick-a-dee calls of chickadees and titmice contain a number of different note types that allow for flexibility in the use of the call. The number of D notes in a call is thought to encode predator attributes, such as size [37-38]-. Quank calls and the number of quank notes are also thought to be associated with increased excitement or potentially predator risk [36], but see [39]. 

We presented mixed-species flocks with either an acoustic cue of predator presence (playback of chickadee chick-a-dee mobbing calls) or a visual cue of predator presence (taxidermy mount of a Northern saw-whet owl, Aegolius acadicus). Predator-related stimuli were presented alone or with playback of traffic noise. If masking is primarily responsible for changes in anti-predator behavior, we would predict a decrease in the response to stimuli presented in the acoustic modality (mobbing calls), but not visual modality (the taxidermy mount). If distraction is primarily responsible for changes in anti-predator behavior, we would predict a decrease in the response to predator-related stimuli presented in both sensory modalities. We used a two-by-two counterbalanced repeated measures design to test these hypotheses. Each trial consisted of a pre-stimulus period to establish a behavioral baseline, a during-stimulus period to measure the effect of the independent variables, and a post stimulus period to capture extended effects of the presented cue (hereafter: pre, during, and post). We used the number of approaches to the stimulus, as well as the anti-predator vocal responses (number of calls and number of D or quank notes, [35-37]) of the three species as response variables.

Methods:

Specifics related to when and where the study was done, including coordinates, would be best presented at the beginning of the methods section to help situate the reader and as is common practice.

Response: We have moved most of the first paragraph of the methods (including the general experimental design) to the introduction as the reviewer has requested. 

Lines 123–133: Revise as goals should be emphasized in the introduction, while your methods should focus on what was done and how. Consider using a more concise way of relating methods to goals: To determine the effects of X on Y, we did Z. Additionally, predictions should be in the introduction and not the methods section (i.e., 127–129).

Response: We have moved this material to the introduction as requested. 

Overall, the methods are scientifically sound and creative! Consider streamlining the methods section to focus on key aspects of the experimental setup and analysis, and relocate the more specific details to the supplementary materials.

Response: Thank you for your appreciation of our methodological approach. We have removed some of the extraneous information about note types that we measured, but did not have sufficient sample sizes to analyze to reduce the length of the methods. We’ve also attempted to streamline the statistics section and standardize some of the language. 

Results: Consider re-organizing and focusing on key findings. I would suggest re-evaluating the necessity of all included figures, and consider moving some of them, along with certain methodological details, to the supplementary materials. This could include data that supports but does not directly contribute to the main narrative of your manuscript.

Response: As suggested, we have focused the results to try to highlight the key findings and how they relate to our hypotheses by adding additional text throughout. We hope this will make the results easier to digest. As for moving some details to the supplementary materials, we have seriously considered the suggestion. However, we feel that our analysis of each of these factors (approaches, number of calls, and number of notes) are each addressing our hypotheses. Selecting only significant results or some of the data would be cherry-picking the results to support our claim. We feel that it is important to document both the significant and not significant findings for each of the response variables to be transparent in our data reporting and for the reader to accurately evaluate the results. Furthermore, neither the second reviewer nor the editor have made any suggestions for modifications to the results, so we would prefer to leave them as is. If the reviewer has specific items that they would like removed or moved to the ESM, we would be happy to consider them on a case-by-case basis. 

Discussion: I believe that after reviewing the manuscript to emphasize important findings, this section will have a clearer narrative. In its current state, it is difficult to relate to directly relate the discussion to the research questions and to the broader literature.

Response: Hopefully the changes we have made to the methods and results address this point. Therefore, we have not made any changes to the discussion. 

Citing: Throughout the manuscript, the authors indicate whether references were the topic was reviewed. In these instances, I suggest they opt for a more concise way of referencing these as to avoid continuously interrupting the flow of the text. It could be replaced by “see” as it is common practice. Additionally, instead of including a long list of citations, particularly in the introduction, consider only using a few representative citations as examples: (e.g., REF1, REF2, REF3).

Response: We have removed the “reviewed in” as the reviewer has suggested. The references will be replaced by numbers in the final draft, so the number of references interrupting the flow should be a non-issue. 

Reviewer #2: The manuscript titled ‘Mixed support for the effects of masking and distraction on anti-predator behavior in suburban anthropogenic noise’ is a well written with proper introduction and discussion. However, the methods part is not up to the mark. There are many loop holes which makes it hard to follow. I have listed my comments as follows

Response: Thank you for your comments. We aim for a thorough and transparent methods section, so we’re happy to provide more information / explanation. It is possible that our familiarity with the subject area led to some inadvertent omissions. We do note that this review of the methods is at odds with the suggestions of reviewer 1 that we reduce the methods section and move much of it to the ESM. We hope that some additional information, as well as some streamlining in two sections will address the suggestions of both reviewers. 

Major comments

Line 112-114: ‘we presented mixed-species……presence (playback of chickadee chick-a-dee mobbing calls)’. Authors have not mentioned the rationale behind using chickadee mobbing calls as acoustic cues? Do all three species always occur together? I also wonder why the call of predator (Norther saw-whet owl) was not used as acoustic cues. A general background of relation between Chickadee, Tuftmice and Nuthatch is missing in the introduction part.

I suggest authors to add justification on using chick-a-dee call of Chickadee and also add a brief about all three species.

Response: Agreed – while this is well known to folks that work with these three species, some context is missing for the broader audience. We have included additional information in the introduction on these three species. We now note that they form mixed-species folks, mob together, and all respond to chickadee mobbing calls at the end of the introduction. Therefore, this should be a salient cue of risk for all three species. The owls are nocturnal and therefore primarily vocalize at night. This would be an incongruous predator cue, which is why we did not use it. We frequently find them together, but they are not always in mixed species flocks. Presumably all of our individuals have interacted with the other species regularly at our study location. The relevant section now reads:

These three species form mixed species flocks during the non-breeding season, engage in predator mobbing together, and are known to respond to the mobbing calls of chickadees [34-35]. Chickadees and titmice both produce chick-a-dee calls and nuthatches produce quank calls in anti-predator contexts [35-37]. The chick-a-dee calls of chickadees and titmice contain a number of different note types that allow for flexibility in the use of the call. The number of D notes in a call is thought to encode predator attributes, such as size [37-38]-. Quank calls and the number of quank notes are also thought to be associated with increased excitement or potentially predator risk [36], but see [39].

In Line no 153-154: It is mention that Northern saw-whet owls are viewed as high threat predators by black-capped chickadees and Chickadees produce vocalizations that encode a high degree of threat to saw-whet owls. However, in line 166-167 it is mentioned that 'The acoustic cue of predator presence consist of chickadee chick-a-dee calls with 3-4D notes that were arrange in 22 or 23 bouts'. Further, in line 172 -173, it states 'Calls of this type have previously been shown to communicate a moderate level of threat’. Thus, statements are confusing and therefore I feel that the chick-a-dee call used as stimulus is not appropriate. It is not clear why authors used a moderate level threat call instead of high-level threat. Justification needed.

Response (circa line 186): The total threat comes from both the type of predator (of which a saw-whet owl is a relatively high threat) and the context (is the predator flying, sleeping, time of day, etc). The predator plus the context should be of similar threat. The number of D notes in the calls we used was in line with the number produced to a saw-whet owl in previous studies (e.g. Templeton et al. 2005 found approximately 4 D notes per call towards saw-whets). I think the confusion is primarily from our phrasing, so we’ve rephrased to make it clearer that these two stimuli align (although a predator mount is still a more concrete risk than a mobbing call). The relevant section now reads:

Calls of this type are in line with those produced to Northern saw-whet owls in previous work [37,44] and all three target species respond to these calls [34].

Line 206-211: Authors have mentioned that they have calibrated the amplitude of chick-a-dee calls and anthropogenic noise to 75 dBA at 1 m. However, the natural amplitude of chick-a-dee call is not mentioned. I suggest authors to add the natural amplitude of chick-a-dee call of chickadee.

Response (circa line 222): We have added that this is in line with normal vocal amplitudes. Chickadees have been shown to dynamically alter the amplitude of their vocalizations (e.g. LaZerte et al. 2016), as well as the frequency and duration of mobbing calls (Courter et al. 2020), so there is no single amplitude or call that will be representative of all vocalizations. Moreover, the amplitude of the call at the receiver will depend on the distance from the call. That being said, this amplitude is in line with our previous measurements and sounds “right” to the human ear as well. The relevant line now reads:

We calibrated the amplitude of the chick-a-dee calls and anthropogenic noise to 75 dBA at 1 m with a Pyle PSPL05R sound level meter (fast integration time) using the previously described white noise calibration stimulus, in line with amplitudes we have recorded at our field sites.

Line 223-224: ‘This tripod was set to a……this height’, since the study focus on three species it would be better if authors could add the foraging height of other two species as well.

Response (circa line 239): Thanks for the note. This was intended to mean that the acoustic stimulus (chickadees) and the predator stimulus (owls) would be appropriate at this height, but that’s not what we wrote! We adjusted this to make it clearer (chickadees call from, and owls perch at, this height). The relevant line now reads:

This tripod was set to a height of 1.7 m, as Northern saw-whet owls are known to hunt from this height [45] and we often observe chickadees vocalizing from this height.

Line 266: Authors claimed that they have counted number of songs and mobbing calls, however, same is not mentioned in result part.

Line 267: Authors also claimed that they have counted number of gargles made by chickadees and the number of high zees made by titmice. Though they have also mentioned that statistical analyses was not performed, I suggest that it will be better to add the result of this part as well. In addition, it will give reader more sense of the study if authors could provide information on what gargles and zees are. 

Response (circa 281-289): These were only found in a very small number of samples (<10% of all recordings), which is insufficient for any statistical analysis. These vocalizations would often be found only during one of the three trial periods, which doesn’t work for our repeated measures design. We’ve opted to remove these sentences to avoid confusion. The same is true of songs, so we’ve removed that as well. Hopefully this will address any confusion. 

Line 272: ‘Finally, we counted……………………. chick-a-dee calls’, this statement is confusing as authors haven’t mentioned before that both chickadee and titmouse both produce chick-a-dee call. One would misunderstand that chick-a-dee is only produce by chickadee. Thus, to avoid confusion, I suggest to add briefly about the calls of three species.

Response (last paragraph of intro): We have added to the introduction that both species produce chick-a-dee calls. The relevant section now reads:

These three species form mixed species flocks during the non-breeding season, engage in predator mobbing together, and are known to respond to the mobbing calls of chickadees [34-35]. Chickadees and titmice both produce chick-a-dee calls and nuthatches produce quank calls in anti-predator contexts [35-37]. The chick-a-dee calls of chickadees and titmice contain a number of different note types that allow for flexibility in the use of the call. The number of D notes in a call is thought to encode predator attributes, such as size [37-38]-. Quank calls and the number of quank notes are also thought to be associated with increased excitement or potentially predator risk [36], but see [39].

Line 272-274: Authors haven’t mentioned what they speculating by counting the number of each note types in chickadee and titmouse chick-a-dee calls and the number of quank notes in nuthatch quank call. I suggest authors to add what they are speculating by counting number of each note types and why it is important. Furthermore, according to Bartmess-LeVasseur et al.2010, no effect predator context in associated with quank call of Nutchatch. So how do you justify this statement?

Bartmess-LeVasseur, J., Branch, C. L., Browning, S. A., Owens, J. L., & Freeberg, T. M. (2010). Predator stimuli and calling behavior of Carolina chickadees (Poecile carolinensis), tufted titmice (Baeolophus bicolor), and white-breasted nuthatches (Sitta carolinensis). Behavioral Ecology and Sociobiology, 64, 1187-1198.

Response: We’ve added a bit to the introduction (last paragraph) and discussion (circa line 532) to address this. The relevant sections now read:

[Introduction] We presented mixed-species flocks with either an acoustic cue of predator presence (playback of chickadee chick-a-dee mobbing calls) or a visual cue of predator presence (taxidermy mount of a Northern saw-whet owl, Aegolius acadicus). Predator-related stimuli were presented alone or with playback of traffic noise. If masking is primarily responsible for changes in anti-predator behavior, we would predict a decrease in the response to stimuli presented in the acoustic modality (mobbing calls), but not visual modality (the taxidermy mount). If distraction is primarily responsible for changes in anti-predator behavior, we would predict a decrease in the response to predator-related stimuli presented in both sensory modalities. We used a two-by-two counterbalanced repeated measures design to test these hypotheses. Each trial consisted of a pre-stimulus period to establish a behavioral baseline, a during-stimulus period to measure the effect of the independent variables, and a post stimulus period to capture extended effects of the presented cue (hereafter: pre, during, and post). We used the number of approaches to the stimulus, as well as anti-predator vocal responses (number of calls and number of D or quank notes, [35-37]) of the three species as response variables.

[Discussion] However, it is also important to note that while quank calls have been implicated in anti-predator behavior [35-36], at least one study as shown that nuthatches did not produce more vocalizations to painted plastic hawk models than they did to similar dove models [39]. 

As for the nuthatches, quanks have been shown to be produced in mobbing contexts in other studies and typically correlate with the intensity of mobbing by other species (e.g. Nolen and Lucas, 2009). However, quank calls have also been shown to more generally be associated with “excitement” and appeared to be given in response to seeing owls and other predators (Long 1982, Ritchison 1983). They also appear to be produced more when nuthatches are together or when humans approach (see birds of the world account). Red-breasted nuthatches do also appear to produce calls in an anti-predator context. That was our approach going in, as a number of studies suggest that they are given in a predator context. Our results may actually support Barmess-LeVasseur et al. 2010, as we didn’t find any main effect of predator on the nuthatches (although predator modality was included in some interactions). We’ve added a note to the discussion that an alternative interpretation is that the quank call may actually be indicating some level of arousal when produced in mobbing contexts, rather than a response to the predator per se. One possible explanation is that the calls are given more in conjunction with predators at nest sites, rather than towards adult predators (there is some hint of this in the literature). 

Grubb Jr., T. C. and V. V. Pravosudov (2020). White-breasted Nuthatch (Sitta carolinensis), version 1.0. In Birds of the World (A. F. Poole, Editor). Cornell Lab of Ornithology, Ithaca, NY, USA. https://doi.org/10.2173/bow.whbnut.01

Richison, G. (1983). Vocalizations of the white-breasted nuthatch. Wilson Bull. 95: 440-451. 

Line 284: Figure 1 do not give clear vision of different notes of chick-a-dee call produce by bot chickadee and titmouse. I suggest authors to add a better spectrogram showing different note types of all three species.

Response: We’ve removed most of the methods corresponding to measuring the different note types and only left the section on D and quank notes. This spectrogram should effectively show those notes for each of the three species. 

Line 293: ‘First, we ………………… species’, Since, experiment was conducted in wild habitat, how authors have controlled the abundance of each species during the experiment? Unequal population size of three species will lead to false estimation of the number approaches of three target species. Therefor, I suggest authors to provide information about the same.

Response: Chickadees were most abundant, followed by titmice, then nuthatches. This is almost certainly what the main effect of species is indicating (we now state this explicitly in the results, circa line 374). However, we were looking for patterns across the trial periods. Thus, each pre-trial period acts as the baseline against which the subsequent trial period can be compared. The exact number of individuals should not affect this (species X trial interactions). If we analyze each species separately, we find the same patterns. 

Line 295-296: ‘we analyzed …………………….. and titmice’. Authors haven’t mentioned why they analysed D note per trial period, why not other notes. As the large audience do not have the background of chick-a-dee calls, it is not easy to follow the statement. I suggest authors to add the importance of D note in chick-a-dee call and how change in number of D notes changes the information content of the call.

Response: We have added additional context to the introduction. The relevant lines now read:

The chick-a-dee calls of chickadees and titmice contain a number of different note types that allow for flexibility in the use of the call. The number of D notes in a call is thought to encode predator attributes, such as size [37-38]-. Quank calls and the number of quank notes are also thought to be associated with increased excitement or potentially predator risk [36], but see [39]. 

Line 314-315: We analyzed ..……... one model. Though authors have normalised the uneven distribution of time which is mentioned in Supporting Information ESM2. I suggest authors to mentioned the same in main text. Same goes to line 325 -326.

Response: We noted in the statistics section that we normalized these (to approach/calls per 10 minutes) on lines 334 and 346. We have added the same to notes (line 349). We have also added a statement that we converted to responses per 10 minutes at the beginning of the stats section (circa line 315). The relevant lines now read:

Approaches, number of calls and number of notes were analyzed as the number per 10-minute interval. The number of notes in the pre and post period values were multiplied by 2 for consistency with the during period.

Line 314-315: Authors have mentioned that they have normalised the time period by multiplying by 2 (also mentioned in Supporting Information ESM2) in the case of pre and post-trials. However, I speculate that number of approaches should decrease by time, for instance, if total number of individual present is 10 and 7 individual approaches in 5 minutes then number of individuals approached in next 5 minutes will be 3 not 7. Therefore, I suggest authors to instead of analysing for 10 minutes, analysing approach for 5 minutes will give better representation of data.

Response: Unfortunately, we do not have the ability to do this, as approaches were measured by tally marks per period in the field. It would be possible to do this for calls and notes (although it would be a significant amount of work that would be unlikely to radically change the outcome of the results). Previous work (e.g. Nolen and Lucas) and my observations in the field is that it can sometimes take a few minutes for birds to detect the predator mount, if they are not facing towards it. It can also take some time for individuals that are not on the feeding platform to be recruited to the area, so I think you would miss things that are occurring in the 10 minute period. We also found persistent effects into the post-stimulus trial period, suggesting 10 minutes is an appropriate time period for our work. This is also a common time frame for these types of experiments (see for instance, the Bartmess-LeVasseur paper the reviewer mentioned above). On the other hand, the pre-trial period represents a random sample of activity and we expect this to be largely representative of activity occurring at that time. We could divide the 10 minute period by 2, but this would be equivalent to the approach we took. As the review notes, it is speculation that this approach would be superior. Moreover, our approach was sufficient to capture differences in call production across time periods (at least for chickadees and titmice). 

Line 548: ‘Therefore, titmice ………….. amplitude noise’. To support this statement, authors should provide the frequency range of noise which will help in understanding the range of frequency masking. If the frequency overlap between noise and chick-a-dee call is <2 kHz, then I suspect that Titmice responding is not just because of greater sensitivity at high frequencies as sensitivity range of chickadee is 2-4 kHz (Wong & Gall 2015).

Response: This is the averaged power spectrum of our five anthropogenic noise files (shown from 0-6kHz, low-pass filtered at 10 kHz and set to 75dB). The peak energy is between 0 and 1 kHz, with a roll off of approximately 9dB/octave, which we have now added to the methods section (circa line 195). Note the dB here refer to the dB of the voltage of the stimulus, but the shape and roll-off should match amplitude measurements. While the peak is certainly below 1 kHz, there is still substantial energy between 2-4 kHz. Some recent work of ours on critical ratios also suggests that critical ratios of the titmice are better than the chickadees at these frequencies, which further supports this idea (although these data are still unpublished, so we do not reference them here. 

Line 52: “masking, distraction, anthropogenic noise, anti-predator behavior”, These words are already present in the title. Key words should be different from the words present in the title.

Response: Keywords updated. 

Line 116: “anthropogenic traffic noise”, there is no natural traffic noise. It can be change to traffic noise.

Response: Changed as requested. 

Line 138: “Platforms were at least 0.4 km….”, it is not clear what authors refer platform to. Is it equivalent to territory size? If so, what is the territory range of these species?

Response: We explained what the platforms were on the next line. We agree that this is confusing, so we have switched the order of those sentences. They now read:

Each location had an elevated feeding platform which was baited with black oil sunflower seeds twice per week starting approximately two weeks before the beginning of data collection and continuing throughout the data collection period to encourage consistent bird activity (as in [18]). Platforms were at least 0.4 km apart and during our ten years of banding and subsequent observations we have never encountered individuals at more than one platform location in a given year.

Line 222: ‘we placed a tripod 5m’, may change to ‘we placed a tripod at a distance of 5 m’

Response: changed as requested. 

Line 486: ‘Number of Notes Per Call’ may replace it with ‘Number of D Notes Per Call’

Response: This section includes an analysis of quank notes per call, so should not be changed as requested by the reviewer. 

Line 79: “(Damsky & Gall, 2017, Templeton et al. 2016)”, put reference in chronological order

Line 90: “Dooling and Blumenrath” change ‘and’ to ‘&’Line 536: ‘(Damsky and Gall, 2017)’ change it to (Damsky & Gall, 2017)

Line 546 & 555: ‘(Henry et al. 2017)’ change it to ‘(Henry et al., 2017)’

Line 552: ‘(Henry & Lucas 2010, Henry & Lucas 2011)’ add ‘,’ after ‘Lucas’

Line 565: ‘(Henry et al. 2011)’ add ‘,’ after ‘al.’

Line 577 &585: ‘Chan et al. 2010’ add ‘,’ after ‘al.’

Line 649: ‘(Sweet et al. 2021)’ add ‘,’ after ‘al.’

Response: Reference format updated to Vancouver style.

---

## [Decision Letter · Decision Letter 1]

4 Aug 2023

Interspecific differences in the effects of masking and distraction on anti-predator behavior in suburban anthropogenic noise

PONE-D-23-10182R1

Dear Dr. Gall,

We’re pleased to inform you that your manuscript has been judged scientifically suitable for publication and will be formally accepted for publication once it meets all outstanding technical requirements.

Kind regards,

Vivek Nityananda

Academic Editor

PLOS ONE

Additional Editor Comments (optional):

Reviewers' comments:

Reviewer's Responses to Questions

**Comments to the Author**

1. If the authors have adequately addressed your comments raised in a previous round of review and you feel that this manuscript is now acceptable for publication, you may indicate that here to bypass the “Comments to the Author” section, enter your conflict of interest statement in the “Confidential to Editor” section, and submit your "Accept" recommendation.

Reviewer #2: All comments have been addressed

2. Is the manuscript technically sound, and do the data support the conclusions?

Reviewer #2: Yes

3. Has the statistical analysis been performed appropriately and rigorously? 

Reviewer #2: Yes

4. Have the authors made all data underlying the findings in their manuscript fully available?

Reviewer #2: Yes

5. Is the manuscript presented in an intelligible fashion and written in standard English?

Reviewer #2: Yes

6. Review Comments to the Author

Reviewer #2: (No Response)

7. PLOS authors have the option to publish the peer review history of their article (what does this mean?). If published, this will include your full peer review and any attached files.

Reviewer #2: No

---

## [Editor Report · Acceptance letter]

10 Aug 2023

PONE-D-23-10182R1 

Interspecific differences in the effects of masking and distraction on anti-predator behavior in suburban anthropogenic noise. 

Dear Dr. Gall:

I'm pleased to inform you that your manuscript has been deemed suitable for publication in PLOS ONE. Congratulations! Your manuscript is now with our production department. 

Kind regards, 

on behalf of

Dr. Vivek Nityananda 

Academic Editor

PLOS ONE